# Patterns and drivers of diatom diversity and abundance in the global ocean

Juan J. Pierella Karlusich ●[1,2,3,4] ✉, Karen Cosnier ●[1,2], Lucie Zinger ●[1,2,35], Nicolas Henry ●[2,5], Charlotte Nef ●[1,2], Guillaume Bernard[1,2], Eleonora Scalco ●[6], Etienne Dvorak[1,2], Tara Oceans Coordinators*, Fabio Rocha Jimenez Vieira[1,2], Erwan Delage[7], Samuel Chaffron ●[2,7], Sergey Ovchinnikov ●[4,8], Adriana Zingone ●[6] & Chris Bowler ●[1,2] ✉

Diatoms constitute one of the most diverse and ecologically important phytoplankton groups, yet their large-scale diversity patterns and drivers of abundance are unclear due to limited observations. Here, we utilize *Tara* Oceans molecular and morphological data, spanning pole to pole, to describe marine diatom diversity, abundance, and environmental adaptation and acclimation strategies. The dominance of diatoms among phytoplankton in terms of relative abundance and diversity is confirmed, and the most prevalent genera are *Chaetoceros*, *Thalassiosira*, *Actinocyclus* and *Pseudo-nitzschia*. We define 25 distinct diatom communities with varying environmental preferences illustrative of different life strategies. The Arctic Ocean stands out as a diatom hotspot with 6 of the diatom communities being exclusive to it. Light harvesting and photoprotection are among the cellular functions in which natural diatom populations invest the bulk of their transcriptional efforts. This comprehensive study sheds light on marine diatom distributions, offering insights to assess impacts of global change and oceanic anthropogenic impacts.

Diatoms are the most abundant and diverse group of eukaryotic phytoplankton, represented by tens of thousands of species[1–3]. Diatoms cover a broad size spectrum, which ranges over more than nine orders of magnitude in cell volume and is further expanded by many chain-forming species[4]. They are believed to be responsible for around 20% of the total primary production on the planet, thus constituting the base of aquatic food webs and contributing to the export of carbon from the surface ocean to depth in many marine environments[5–7]. Their cells possess a rigid silica cell wall, denoted frustule, which makes them the world's largest contributors to biosilification[8]. They are also important players in the nitrogen cycle because they have efficient

strategies for nitrogen capture and utilization[9,10]. Furthermore, some species even harbour nitrogen-fixing symbionts[11–14]. Given their important role in biogeochemical cycles, food webs and carbon export, understanding what drives diatom diversity and spatial distribution is an important topic, in order to better predict the responses and resilience of marine ecosystems to natural and human-induced perturbations.

Field observations and modelling efforts have shown that marine diatoms are especially common in well-mixed coastal and upwelling regions and at high latitudes[2,15,16]. However, how different diatom populations (defined here broadly in terms of species, genera, size

[1]Institut de Biologie de l'Ecole Normale Supérieure (IBENS), Ecole Normale Supérieure, CNRS, INSERM, Université PSL, Paris, France. [2]CNRS Research Federation for the study of Global Ocean Systems Ecology and Evolution, FR2022/Tara Oceans GOSEE, Paris, France. [3]FAS Division of Science, Harvard University, Cambridge, MA, USA. [4]Department of Biology, Massachusetts Institute of Technology, Cambridge, MA, USA. [5]CNRS, FR2424, ABiMS, Station Biologique de Roscoff, Sorbonne Université, Roscoff, France. [6]Stazione Zoologica Anton Dohrn, Villa Comunale, Naples, Italy. [7]Nantes Université, CNRS UMR 6004, LS2N, Nantes, France. [8]John Harvard Distinguished Science Fellowship Program, Harvard University, Cambridge, MA, USA. [35]Present address: Centre de Recherche sur la Biodiversité et l'Environnement (CRBE), CNRS, Université Toulouse III, IRD, INP, Toulouse, France.*A list of authors and their affiliations appears at the end of the paper. ✉e-mail: pierella@mit.edu; cbowler@bio.ens.psl.eu

classes, or morphotypes) diverge in their biogeography and niches at the global ocean scale is still hindered by limited observations and methodological inconsistencies. For example, while satellite remote sensing now allows to estimate the distribution of diatoms and other phytoplankton coarse groups based on optical properties at the sea surface, it fails to discriminate their taxonomy, to detect subsurface blooms or to determine variable cellular pigment quotas[17–19]. The Continuous Plankton Recorder (CPR) survey has been collecting plankton records since the 1930s but is mainly useful for quantifying larger-celled species[20]. The first global ocean database for diatoms was compiled for the MARine Ecosystem DATa (MAREDAT) project, mainly using classical microscopy counts from bottle samples collected in regional studies[21]. Over the last years, genetic surveys have been producing an impressive amount of new data about diatom distributions. These investigations are typically based on PCR amplification and sequencing of a fragment of the small subunit of the rRNA gene from the DNA isolated from an environmental sample (a method hereafter referred to as metabarcoding[22]. rRNA gene-based metabarcoding has a comparatively deeper taxonomic resolution (including for cryptic species) as well as higher detection power (e.g., for rare species) than routine optical methods[23–25]. Currently, the only available large-scale and standardized molecular survey of marine planktonic diatoms was carried out during the first leg of the *Tara* Oceans expedition (46 sampling sites) by targeting the V9 fragment of the 18S rRNA gene[2]. This study was able to improve diatom diversity estimates, e.g., 43% of the observed genera were not represented in the MAREDAT database. However, there was a poor representation of areas expected to have high diatom abundances, such as polar regions.

These taxonomic surveys can be complemented by metagenomic and metatranscriptomic approaches to infer diatom functional diversity, as well as their adaptation and acclimation mechanisms (i.e., variations in gene and transcript copy numbers, respectively) across environmental gradients. The ecological success of diatoms depends on a range of adaptive attributes in addition to the frustule, such as prominent vacuoles for nutrient storage, ice-binding proteins to withstand harsh polar environments, proton pumping rhodopsins to convert light energy into chemical energy in low-light environments, and a metazoan-like urea cycle for cellular management of nitrogen[9,26–28]. However, between 30 and 60% of diatom genes still have unknown or poorly described functions[29]. In addition, functional studies of diatom populations have been restricted to specific genera in local and regional studies[30–35]. Therefore, many gaps still remain in understanding the molecular mechanisms underpinning diatom success and their distribution in the contemporary ocean.

In this work, we aim to provide a pole-to-pole view of the diversity, adaptation and acclimation features in natural populations of diatoms. This study expands the previous survey by Malviya et al.[2] from *Tara* Oceans not only by covering a much higher number of sites (from 43 to 144 stations) in all ocean regions but also by the significant incorporation of two new regions: the North Pacific Ocean (NPO) and the Arctic Ocean (AO). We also provide a robust picture of diatom taxonomic diversity and distribution through the comparative analysis of DNA data using two DNA markers (i.e., the V9 and V4 variable regions of the 18S rRNA gene; only V9 had been used in the previous study) together with observations obtained through microcopy. Finally, we mine the metagenomes and metatranscriptomes of the same samples to explore the adaptation and acclimation processes that may underlie diatom distribution patterns.

## Results
### Prevalence of diatoms among eukaryotic phytoplankton and silicifiers
To capture the whole size spectrum of plankton, a combination of filter membranes with different pore sizes were used to serially size-fractionate organisms by cell diameter and aggregation forms

(specifically from 0.8–5 μm or 0.8–2000 μm, 3–20 or 5–20 μm, 20–180 μm, and 180–2000 μm)[36]. V4 and V9 amplicons were sequenced from these size-fractionated samples collected in epipelagic waters from 144 stations (Supplementary Fig. S1 and S2). Amplicon sequence variants (ASVs) were inferred using denoising methods (DADA2)[37] and taxonomically assigned using IDTAXA[38] with the PR2 database version 4.14[39] (see Methods). We retrieved ASVs assigned to eukaryotic phytoplankton and normalized their read abundance by the total eukaryotic read abundance of the corresponding sample.

We found that diatoms are overall the most abundant phytoplankton group (Fig. 1a, b and Supplementary Fig. S3a, b). In addition, we found that diatoms exhibited the largest variation in relative abundance, from 0.0002% to 95%, while other phytoplankton groups had abundance variations of up to 62% for photosynthetic dinoflagellates (hereafter dinoflagellates), 38% for haptophytes, 32% for pelagophytes, 13% for dictyochophytes (another silicifying group), and 16% for prasinophytes (Fig. 1a and Supplementary Fig. S3a). Therefore, diatoms constitute an important group due to their dominance and their higher sensitivity to environmental variations, which likely has important consequences for the ecosystem in the context of climate change. Given that diatoms have been used to assess freshwater quality for decades[40], this result also suggests a similar use as indicators for the ocean.

Diatom and dinoflagellate relative abundance values based on 18S rRNA gene metabarcoding were higher compared to those inferred from the single-copy marker gene *psbO* found in metagenomes[41]. This difference could be attributed at least partly to the larger cell size of these organisms. Although cell size is not the only factor influencing cell rRNA copy number[42], larger cells generally require a higher number of copies of the 18S rRNA gene to maintain cell activity[43,44]. Therefore, the prevalence of diatoms among the 18S rRNA gene barcodes could reflect to some extent a higher relative biovolume proportion.

Further, diatoms were also found to be the leading group among silicifiers[45], rivalled only by Nassellaria, an order of polycystine rhizarians with silica-based skeletal structures (Fig. 1d, e and Supplementary Fig. S3c, d). All in all, this survey reaffirms the dominance of diatoms within both phytoplankton and silicifier communities, highlighting their integral role in marine biogeochemistry and their potential as ecological indicators in a changing global climate.

### Diatom ASV diversity
We focused on diatoms by selecting only those V9 and V4 ASVs assigned to this group. From 769 samples and almost 385 million V4 reads in total, we obtained a total of >19.5 million reads and 4,998 ASVs for diatom V4, while from 889 samples and >1554 million V9 reads, we obtained >105 million reads and 3957 ASVs for diatom V9. In comparison, the 293 samples of the initial segment of the *Tara* Oceans expedition analysed in Malviya et al.[2] produced less than half the number (1761) of diatom V9 ASVs.

We obtained a higher number of diatom ASVs for V4, and accumulation curves and richness estimators indicated that both V4 and V9 ASV diversity approached saturation at a global scale (Supplementary Fig. S4 and Supplementary Table S1). The V4 marker is almost three times longer (385 bp *vs* 130 bp for V9), which increases the potential for genuine variation as well as the likelihood of PCR/sequencing errors that may not be fully detected by the DADA2 algorithm[46,47]. It is important to note that the ASVs can account for variation not only at the species level but also at intraspecific and intragenomic levels.

The diatom ASVs represented 34% and 42% of the total V4 and V9 ASVs attributed to phytoplankton, respectively (Fig. 1c). This highlights diatoms as the most diverse group within the global phytoplankton community, followed by dinoflagellates and haptophytes. However, photosynthetic dinoflagellate diversity might have been underestimated because a great number of ASVs cannot be classified

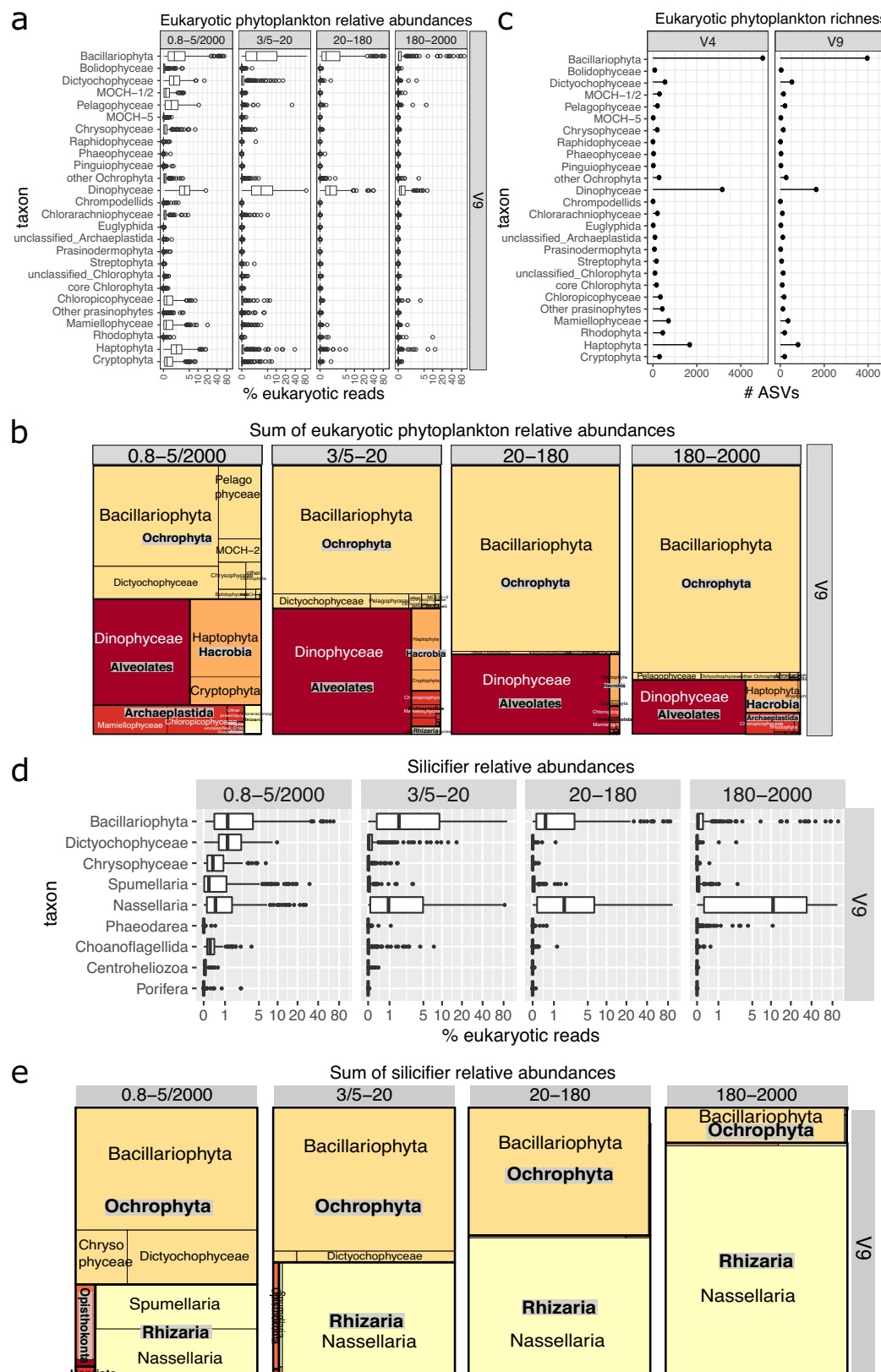

as either photosynthetic or non-photosynthetic (e.g., those categorized as 'unknown dinoflagellate' or 'unknown Gymnodiniales').

## Patterns of diatom abundance and diversity
When examining, diatom reads relative abundance across marine biomes and oceanic regions, the highest values were observed in polar biomes (Supplementary Fig. S5c, d). Accordingly, a relative abundance latitudinal gradient towards the poles is clearly observed in all size classes (Fig. 2a, b and S6a-b). This is consistent with the larger thermal breadths and lower minimal thermal growth of diatoms when compared with cyanobacteria and dinoflagellates[48]. In contrast, diatom diversity follows a latitudinal gradient with a decrease in species

**Fig. 1 | Contribution of diatoms to eukaryotic phytoplankton (a-c) and silicifiers (d-e) in epipelagic waters of the global ocean sampled by *Tara* Oceans using 18S rDNA metabarcoding data obtained from different size-fractions.** **a–c** Phytoplankton. We focused exclusively on the phytoplankton signal of these data sets based on a functional database (https://zenodo.org/record/3768951#. YM4odnUzbuE), including dinoflagellates and chrysophytes, though we acknowledge there are uncertainties in assigning photosynthesis capacity in these groups. **a** Relative abundances for V9 marker (log scale). Each point is a size-fractionated sample. **b** Sum of normalized reads for the V9 marker in the overall dataset. The equivalent panels a and b for the V4 marker are displayed in Fig. S3. **c** ASV richness for both V4 and V9 marker regions in the overall dataset. **d, e** Silicifiers. **a** Relative abundance for V9 marker (log scale). Each point is a size-fractionated sample. **b** Sum of normalized reads for the V9 marker in the overall dataset. The equivalent panels (**a** and **b**) for the V4 marker are displayed in Supplementary Fig. S3. Boxplots illustrate the distribution of the dataset, with the box representing the 25–75% interquartile range and the central line indicating the median (50% quantile). Whiskers extend to data points within 1.5 times the interquartile range. The V9 dataset comprises 212 samples for the 0.8–5 μm or 0.8–2000 μm size fractions, 186 for the 3–20 μm or 5–20 μm fractions, 194 for the 20–180 μm fraction, and 200 for the 180–2000 μm fraction.

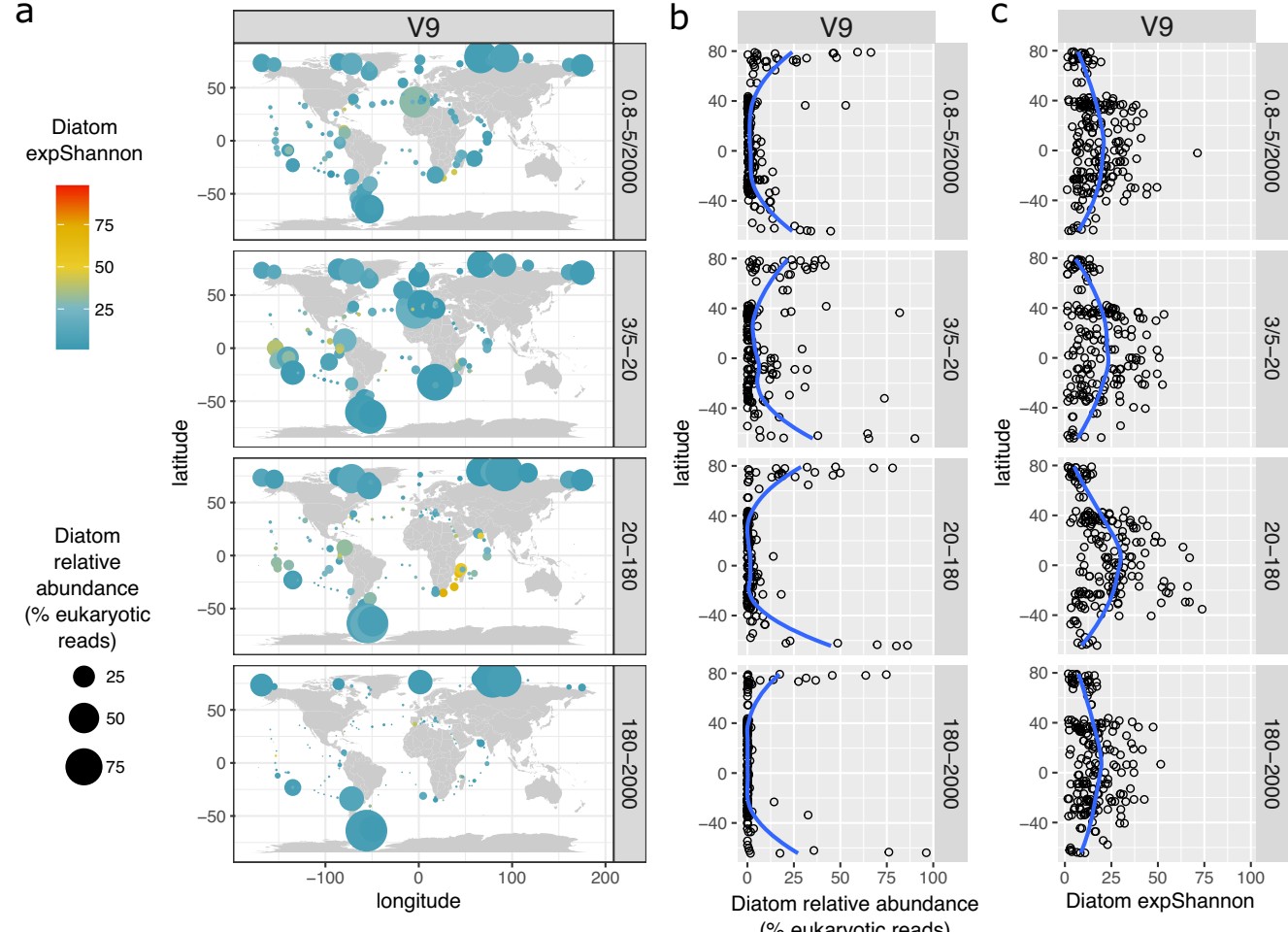

**Fig. 2 | Latitudinal gradients of diatom relative abundance and diversity in surface waters using V9 metabarcoding data obtained from different size-fractions.** **a** Distribution map. **b** Latitudinal gradient for relative abundance. **c** Latitudinal gradient for the exponentiated Shannon Diversity Index. The blue lines correspond to Loess smoothings. The equivalent plots for the V4 marker are displayed in Supplementary Fig. S6. Maps were generated with the *borders()* function in *ggplot2*[89].

richness towards the poles (Fig. 2c and S6c), where diatom communities are likely hyper-dominated by a lower number of species, as observed in previous studies[49].

We employed Partial Least Squares (PLS) and Spearman correlation analyses to explore environmental factors affecting diatom relative abundance and the Shannon diversity index (Fig. 3a and SupplTables S2 and S3). Diatom Shannon diversity and relative abundance generally correlate with the same environmental variables, but in opposite directions, e.g., for temperature (Spearman's rho = − 0.34 and 0.36, respectively; $p < 0.05$). The positive correlation between diatom relative abundance and chlorophyll suggests that diatoms are usually the dominant group in high-biomass

phytoplankton communities (Spearman's rho = 0.47, $p < 0.05$). We also observed a significant positive association between diatom abundance and nitrate concentration (Spearman's rho = 0.43, $p < 0.05$), underscoring nitrate availability as a key factor for diatom growth. In addition, diatom abundance tends to moderately increase with phosphate (Spearman's rho = 0.35, $p < 0.05$) and silica (Spearman's rho = 0.25, $p < 0.05$), while decreasing with the ratio of ammonia to total dissolved inorganic nitrogen (DIN; e.g., sum of ammonia, nitrite, and nitrate) (Spearman's rho = − 0.37, $p < 0.05$). The role of biotic interactions is also significant. Diatoms show a negative association with the picocyanobacterium *Prochlorococcus* (Spearman's rho = − 0.39, $p < 0.05$), supporting findings that diatoms

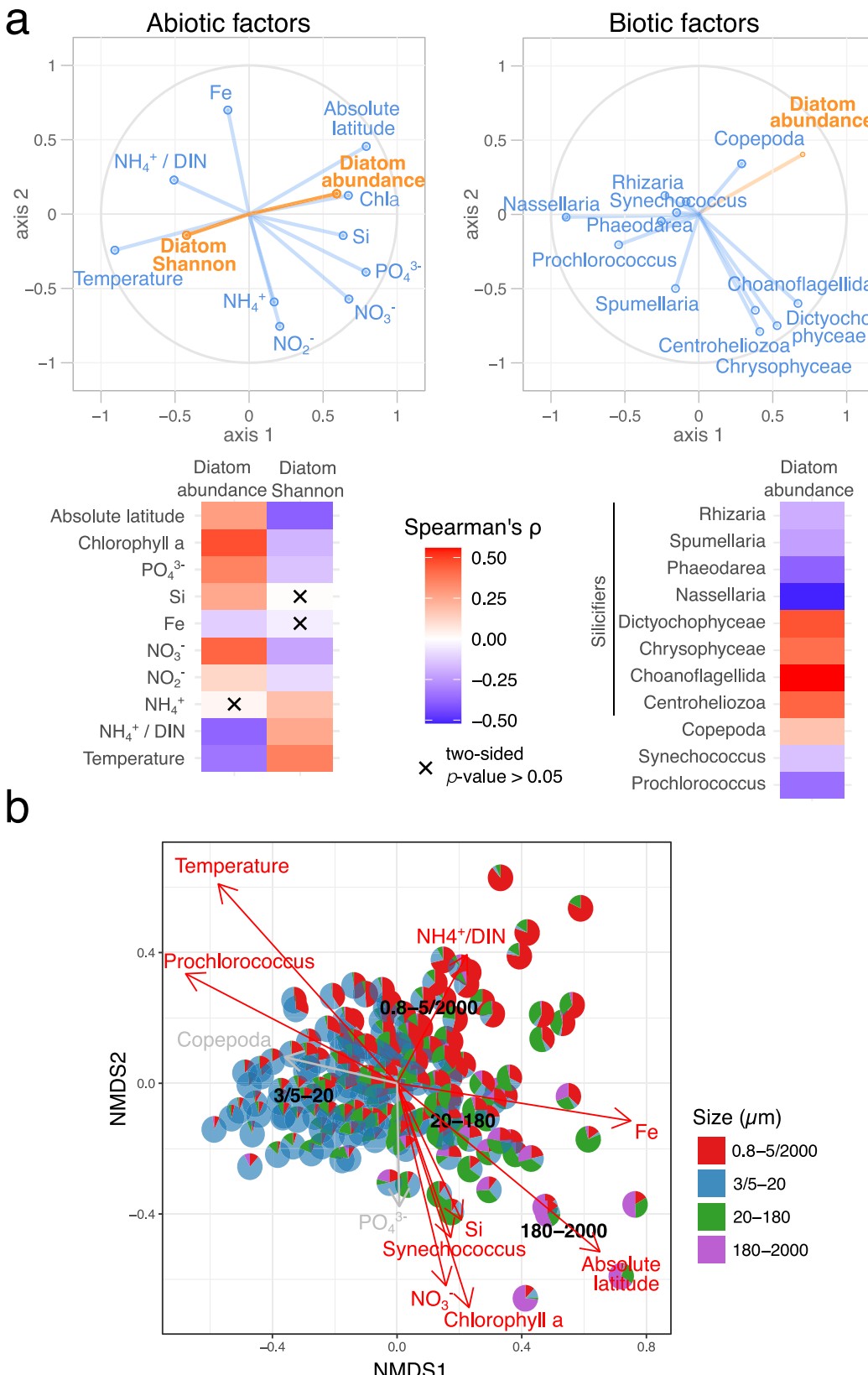

can be outcompeted in nitrogen-limited environments where cyanobacteria efficiently utilize ammonia[50,51], often leading to low chlorophyll levels. In contrast, grazing organisms such as copepods exhibit a weak positive relationship with diatoms (Spearman's rho = 0.20, $p < 0.05$), suggesting grazing top-down control of diatom populations. Lastly, the presence of abundant silicified radiolarians (Nassellaria, Spumellaria, and Phaeodarea) negatively correlates with diatom abundance (Spearman's rho = −0.83, −0.22, and −0.34; respectively, $p < 0.05$), likely reflecting competitive interactions for silica utilization[45].

**Fig. 3 | Diatom abundance and diversity in relation to environmental parameters. a** Correlation circle based on the partial least square analysis (top) and heatmap based on Spearman's rho correlation (bottom) of V4 and V9 diatom relative abundance and Shannon diversity as responses and a diverse set of abiotic (left) and biotic (right) variables as predictors: absolute latitude, nutrient concentrations (ammonia, nitrite, nitrate, phosphate, iron, silica), the ratio of ammonia concentration to total dissolved inorganic nitrogen (DIN), temperature, chlorophyll a concentration, grazers (copepods), picocyanobacteria (*Synechococcus* and *Prochlorococcus*), and silicifiers (the radiolarians of the orders *Nassellaria*, *Spumellaria*, and *Phaeodarea*, the ochrophytes *Chrysophyceae* and *Dictyochophyceae*, choanoflagellates, and centroheliozoa). **b** Pattern of the contribution of diatom reads across size fractions in relation to biotic and abiotic factors. Non-metric multidimensional scaling (NMDS) analysis of stations according to Bray–Curtis distance between size assemblages, with fitted statistically significant physico-chemical parameters (black, adjusted p-value < 0.05; grey, 0.05 <adjusted p-value < 0.10). Each pie chart is the V4 or V9 data from a single station. NMDS stress value: 0.1055151.

## Diatom size patterns

As a reflection of their wide size range, diatoms were relatively abundant in the nanoplankton (3/5–20 μm; mean relative abundance of ~9% of eukaryotic reads), but also in the pico nanoplankton (0.8-5/ 2000 μm; ~5% mean abundance) and in the micro- and mesoplankton (20−180 μm and 180−2000 μm; ~5% and 3% mean abundances, respectively) (Fig. 1a, Supplementary S3a and S5a). Historically, the abundance of smaller diatoms has been underestimated due to detection challenges using microscopy, but recent morphological and molecular studies support our observations of their prevalence in various marine regions[52–55]. The broad size spectrum of diatoms is further evidenced by previous research on their cell volumes, spanning from 3 μm³ (*Thalassiosira* sp.) to $4.71 \times 10^9$ μm[4,21]. It is worth noting that we detected in many cases, the same diatom species across all size fractions. Although we cannot discard the possibility of filter clogging or extracellular DNA adsorbed on particles of varying sizes, this can also be attributed to life-cycle variations (e.g., asexually dividing diatoms decreasing in cell size, rare gamete formation), and the physical characteristics of certain species (e.g., long needle-like cells and broken colonies passing through small mesh sizes)[56].

There is a clear environmental partition between diatom sizes, with larger diatoms associated with polar regions rich in nitrate, whereas small diatoms are prevalent in oligotrophic and temperate waters with high ratios of ammonia to DIN (Fig. 3b). This is in line with Bergmann's rule of reduced organism body size in warmer environments[57]. Thus, the energy transfer from diatoms to higher trophic levels may be reduced in temperate environments due to their smaller sizes[58].

## Environmental distribution of diatom classes

Following Medlin and Desdevises 2020[59], diatoms are classified into three monophyletic classes: Coscinodiscophyceae (basal radial centric diatoms), Mediophyceae (polar centric diatoms and radial centric Thalassiosirales), and Bacillariophyceae (pennate diatoms). Pennate diatoms are further divided into raphid (Bacillariophycidae) and araphid genera (i.e., subclasses Urneidiophycida and Fragilariophycidae).

For both V4 and V9, Mediophyceae were the most prevalent, followed by raphid pennates and Coscinodiscophyceae (Fig. 4a and Supplementary Fig. S8a). Araphid pennates were considerably less abundant and only prevalent in an ice edge station in the AO (TARA_188) (Supplementary Fig. S9). Our observation of low abundance of araphid pennates is consistent with their typical presence in benthic rather than planktonic environments[1,60].

Mediophyceae were highly abundant in regions characterized by high chlorophyll concentrations and high latitudes (Fig. 4b, c and Supplementary Figs. S7, S8b, and S9). Similarly, Coscinodiscophyceae were also notably abundant in high-latitude regions. In contrast, raphid pennates were predominantly found in temperate and stable environments, although they were also present in significant numbers at some high-latitude stations. There was also a compositional shift among the different size fractions of plankton. The relative abundance of Mediophyceae was highest among the piconano-, nano-, and microplankton (Fig. 4a and Supplementary Fig. S7a). Although their relative abundance decreased among mesoplankton, Mediophyceae continued to be the dominant diatom class in this size range (Supplementary Fig. S8a). In contrast, the relative abundance of raphid pennate diatoms demonstrated a continuous decline as the size fractions increased. Coscinodiscophyceae exhibited a predominant presence, particularly within the nano- and to a lesser extent, microplankton size fractions, with their numbers decreasing in the piconano- and, particularly, the mesoplankton. All these biogeographical and size patterns are driven by the patterns of the main genera within each class (see below).

## Diatom genera composition

We found a total of 75 genera for V4 barcodes, and 64 genera for V9 barcodes (Fig. 5). The higher number of genera generated by V4 barcodes is probably related to its higher taxonomic resolution and better representation in the reference database (PR2 v4.14[39]).

High variations across genera were observed in terms of abundance and richness. *Chaetoceros* was the most abundant genus, followed by *Thalassiosira*, *Actinocyclus*, and *Pseudo-nitzschia* (Fig. 5; see also next section). Intragenus richness ranged from 571 V9 ASVs and 945 V4 ASVs for *Chaetoceros* to as low as 1 ASV for some other genera (Fig. 5). This result reflects the differences in known species for each genus (e.g., >100 for *Chaetoceros*, a few for *Arcocellulus* and *Lauderia*, and only 1 for *Tenuicylindrus* and *Pseudohimantidium*). Additionally, this result parallels read abundance, indicating a greater potential of capturing intraspecific and intragenomic variations of these multicopy markers, as well as a higher likelihood of PCR/sequencing errors.

We found ~65% of cumulative read relative abundances in surface samples (Fig. 5), which reflects the sampling bias (513 samples from the surface and 267 from DCM; Supplementary Fig. S5b). When only focusing on locations with both SRF and DCM samples, we observed that most genera were more abundant in surface waters (e.g, *Eucampia*, *Thalassiothrix*, *Rhizosolenia*, *Proboscia*) or equally abundant in surface and DCM. The only genus enriched in DCM was the centric diatom *Attheya*, which is restricted to AO waters (Supplementary Fig. S10).

A wide range of genera were identified among the piconano-plankton communities. Besides the expected genera such as *Minidiscus*, *Minutocellus*, *Arcocellulus*, *Brockmanniella*, *Cyclotella*, and *Cocconeis* (Fig. 5), we observed several typically larger genera traditionally classified as microplankton (20-180 μm). These included the centric genera *Hemiaulus*, *Dactyliosolen*, and *Detonula*, as well as the pennate genera *Pseudo-nitzschia*, *Surirella*, *Thalassionema*, and *Triceratium*, which often form chains and are generally larger. Their presence in smaller plankton samples could be due to broken cells, cells passing through the mesh by their shorter dimension, or possibly being gametes.

## Biogeography of the most abundant diatom genera

We focused on the top 20 most abundant genera (Fig. 6a) since they accounted for the majority of the signal from known taxa, with 97% of assigned reads in the entire V4 dataset and 98% using V9. We compared two normalization strategies—normalization by total eukaryotic reads and by total diatom reads—and summed the relative abundances across all size-fractioned samples (Fig. 6a). The normalization choice presents a trade-off: normalizing by eukaryotic reads tended to over-represent taxa abundant in high-latitude regions dominated by

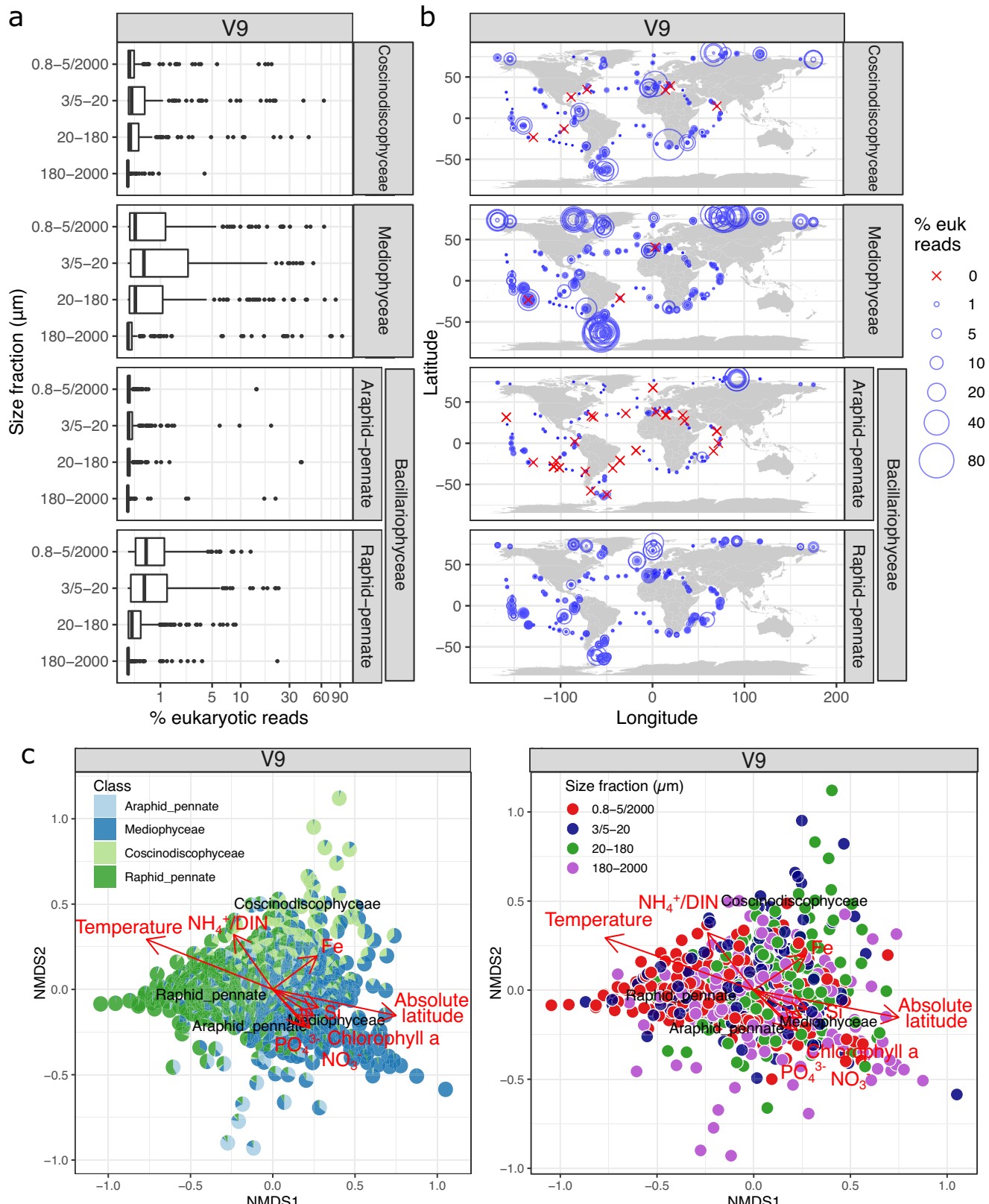

diatoms (*Fragilaria*, *Stellarima*, and *Attheya*); while diatom-based normalization over-represented taxa from temperate waters where overall diatom abundance was lower (*Haslea*, *Coscinodiscus*, *Cyclotella*, and *Pleurosigma*) (Supplementary Figs. S6a and 7). Despite these biases, the majority of genera appear consistently in the top 20 list across both normalization approaches.

Among the top 20 most abundant genera when based on total eukaryotic reads, 15 were consistently found in both V4 and V9 datasets, including *Chaetoceros*, *Thalassiosira*, *Actinocyclus*, *Pseudo-nitzschia*, *Porosira*, *Proboscia*, *Corethron*, *Leptocylindrus*, *Guinardia*, *Eucampia*, *Skeletonema*, *Bacteriastrum*, *Minidiscus*, *Fragilariopsis*, and *Planktoniella* (Fig. 6a). Six genera appear only in one dataset but still

**Fig. 4 | Abundance patterns of diatom classes based on the V9 marker.**
**a** Relative abundance (log scale). Each point is a size-fractionated sample.
**b** Biogeography. Each circle is a size-fractionated sample. **c** NMDS analysis of stations according to Bray−Curtis distance. Fitted statistically significant physicochemical parameters (adjusted *P*-value < 0.05, two-sided test) are displayed: nitrate, phosphate, silicon, iron, ratio of ammonium to total dissolved inorganic nitrogen (DIN), temperature, absolute latitude, and chlorophyll concentration. Each pie chart (left) or each circle (right) is a size-fractionated sample. NMDS stress value: 0.1470158. Comparisons between V4 and V9 patterns are displayed in

Supplementary Fig. S7. The maps separated by diatom classes are shown in Supplementary Fig. S9. Boxplots illustrate the distribution of the dataset, with the box representing the 25−75% interquartile range and the central line indicating the median (50% quantile). Whiskers extend to data points within 1.5 times the interquartile range. The dataset comprises 212 samples for the 0.8−5 μm or 0.8−2000 μm size fractions, 186 for the 3−20 μm or 5−20 μm fractions, 194 for the 20−180 μm fraction, and 200 for the 180−2000 μm fraction. Maps were generated with the *borders()* function in *ggplot2*[89].

rank high in the other (between positions 21 and 29: *Nitzschia, Rhizosolenia, Hemiaulus, Minutocellus, Attheya,* and *Stellarima*), indicating the robustness of our results. Nonetheless, a few differences were observed, with *Fragilaria, Asteromphalus,* and *Brockmanniella* found only in V4, and *Trieres/Odontella* mainly detected in V9 (Fig. 6a). These discrepancies could be due to differences in the coverage and quality of reference databases, the resolving power of the DNA barcode, and variations in PCR amplification efficiency among different taxa. The top 20 genera based exclusively on analysing diatom-specific reads were similarly consistent between V4 and V9 datasets (14 shared genera, and seven top 20 genera only in one dataset but still ranking high in the other) (Fig. 6a).

We found 6 new genera among the V9 top 20 in comparison with the previous *Tara* Oceans V9 survey (see Fig. 6A in Malviya et al.[2]), which did not include AO, NPO and most of NAO (Supplementary Fig. S1b). As expected, the list contains genera prevalent in these new regions: *Bacteriastrum* was abundant in NPO, SPO and IO; *Porosira* and *Skeletonema* were abundant in AO (despite *Skeletonema* being previously reported as absent from polar waters[61]) (Figs. 6a and 7). However, the remaining three genera were more abundant in SO: *Hemiaulus,Trieres/Odontella* and *Stellarima* (Figs. 6a and 7). Although SO was already included in Malviya et al.[2], the higher regional coverage of the present work explains this result for *Trieres/Odontella* (6 current stations *vs* 2 stations in the previous survey), and the updated reference database for *Hemiaulus and Stellarima*.

Given that the 20−180 μm size fraction was also quantified using optical microscopy[2,62,63] (98 samples from surface and/or DCM from 66 stations), we compared the most abundant genera between methods in this size fraction (Fig. 6b). The comparison resulted in a high degree of similarity: 11 of the top 20 genera based on microscopy are also in the top 20 in both molecular datasets, plus six top 20 genera in the microscopy data that are only in one molecular dataset. Some of the differences are explained by the scarcity or lack of references for some genera (e.g., *Cylindrotheca* and *Thalassionema)* or the difficulty of detecting and/or identifying some genera by microscopy *(e.g., Arcocellulus, Minidiscus* and *Haslea)*. A particular case is *Fragilaria*, whose ASVs were primarily found at an Arctic ice station, where the morphologically similar and closely related *Fossulaphycus* (previously named *Fossula*) was abundant in microscopy observations. While *Fragilaria* is considered freshwater, *Fossulaphycus* is typically associated with Arctic sea ice[64]. Therefore, the detected ASVs assigned to *Fragilaria* may actually be *Fossulaphycus*, but we retain the name *Fragilaria* due to the lack of reference sequences of *Fossulaphycus*.

We observed high variability in the biogeography of the most abundant diatom genera (Fig. 7 and Supplementary Fig. S11 and Supplementary Data 1). Some of them, such as *Porosira, Fragilaria, Trieres/Odontella,* and *Attheya*, were regionally constrained, i.e., highly abundant in a few specific stations (Fig. 7 and Supplementary Fig. S11). The top 10 genera, which constituted between 82 and 94% of total diatom reads, tended to be more abundant in high latitudes. There was a group of cosmopolitan genera whose abundances were much higher in polar regions: *Chaetoceros, Thalassiosira, Actinocyclus* and *Fragilariopsis. Pseudo-nitzschia and Nitzschia*, other cosmopolitan genera, were more homogeneously distributed in the different ocean regions. Other genera were mainly detected in polar regions: *Porosira* and

*Proboscia* were abundant in both AO and SO, *Fragilaria* in the AO, *Corethron, Trieres/Odontella,* and *Hemialus* in the SO. Finally, some genera were prevalent in a few stations from tropical and subtropical regions: *Guinardia, Leptocylindrus, Bacteriastrum*, and *Asteromphalus*. The top 11−20 diatom genera, which represented roughly 5−10% of total diatom reads, also included cosmopolitan taxa (*Eucampia, Skeletonema, Rhizosolenia*) and those with more regionally specific distributions (*Attheya* in AO, *Stellarima* in SO), but also a noticeable increase in genera that were abundant in temperate waters (*Cyclotella, Haslea, Planktoniella, Brockmaniella, Coscinodiscus, Minutocellus, Pleurosignma*). Overall, these results confirm the importance of the higher geographical coverage of the present work, especially with the inclusion of AO.

## Communities of co-occurring diatom lineages and their association to environmental gradients

We applied a weighted gene correlation network analysis (WGCNA)[65] to detect co-occurring diatom sub-communities. For both V4 and V9 datasets, we found a total of 25 modules (and a set of unclustered ASVs grouped into the grey module) with similar environmental variables associated to them (Fig. 8 and Supplementary Fig. S11). The modules were clustered according to their co-occurrences into 4 main groups: (i) one formed by 6 modules prevalent in the AO and dominated by Mediophyceae ASVs, (ii) another formed by 4-5 modules mainly detected in the SPO (and to a lesser extent in the NPO) and dominated by Mediophyceae and raphid pennate ASVs, (iii) another group formed by 2-4 modules abundant in the SO/SPO and with Mediophyceae and raphid pennate diatoms, and iv) a group of different basin-specific modules detected in tropical and subtropical regions, including 2 modules prevalent in the IO, 1-2 in the MS, 2 in the NAO (only in V4), and 2 in the SAO (only V9). Therefore, we found diatom sub-communities with a significant association to each oceanic basin or to basins that are physically connected (e.g., the SPO and IO via the Indonesian Passage). The AO stands out from the rest as it contains a relatively high number of AO-specific sub-communities. The reasons why the AO is rich in endemic diatom communities might be related to its particular physical oceanography (the influx from large rivers, and the Atlantic and Pacific Ocean waters), long residence times, and the extreme seasonality in day length and sea ice cover, as well as low temperatures and variable salinity[66].

Overall, *Chaetoceros, Pseudo-nitzschia* and *Thalassiosira*, previously identified as the most prevalent diatom genera, were found within almost all modules, together with unidentified Bacillariophyta, Mediophyceae and raphid pennates (Supplementary Fig. S15).

For both V4 and V9, the turquoise and grey modules gathered the most ASVs, with the grey modules containing a set of unclustered ASVs that were the most enriched in different taxa. Both grey modules were negatively correlated with iron, oxygen and latitude while positively correlated with temperature (Fig. 8). Together with the fact that they harboured diatoms from a diverse range of basins and sizes, the results suggest that the grey modules represent a wide group of rather temperate diatoms; while the absence of other significant correlations of these modules with environmental parameters indicates potential generalist species, or that the wide biological diversity of these modules precludes the identification of a general pattern.

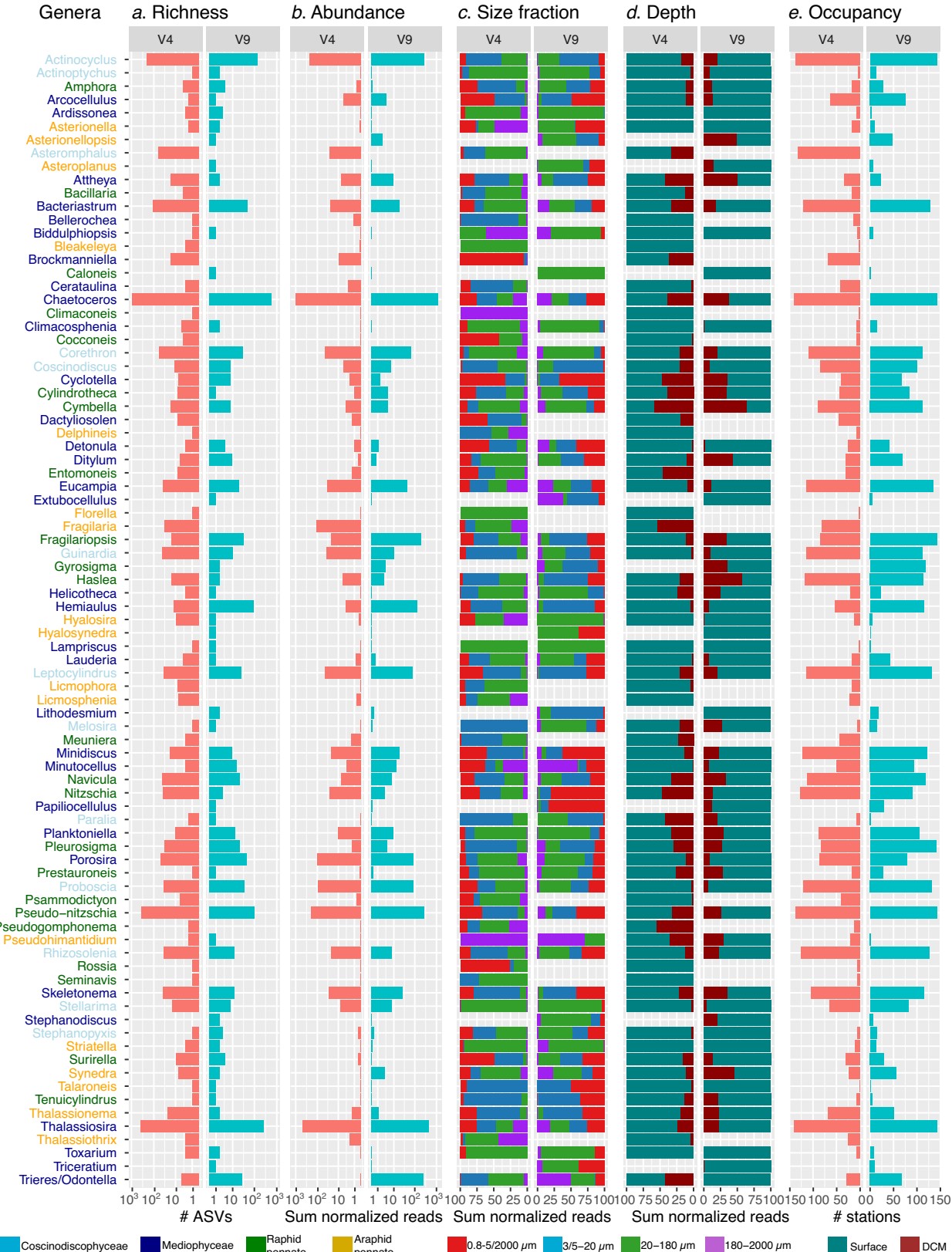

**Fig. 5 | Summary of diatom genera detected in the V9 and V4 metabarcoding datasets.** A total of 75 genera were defined for V4 barcodes and 64 for V9 barcodes. (Column **a**) Richness expressed as the number of ASVs (log scale). (Column **b**) Abundances expressed as the sum across samples of the relative abundances of rDNA reads (log scale). (Column **c**) Percentage distribution of rDNA read relative abundances per size fraction. (Column **d**) Percentage distribution of rDNA read relative abundances per depth. (Column **e**) Occupancy expressed as the number of stations in which the genus was observed.

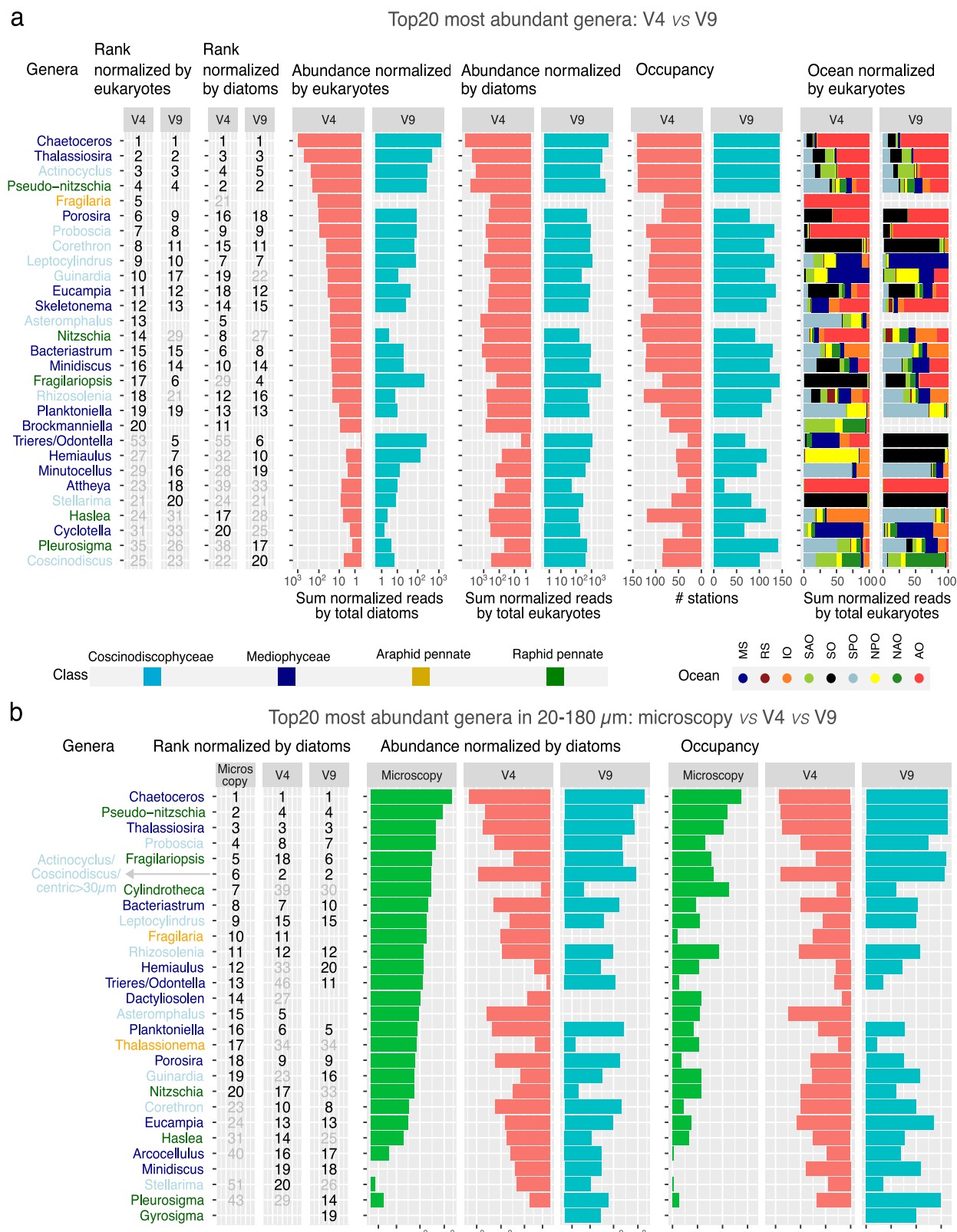

**a** Top20 most abundant genera: V4 *vs* V9

**b** Top20 most abundant genera in 20-180 µm: microscopy *vs* V4 *vs* V9

Exploring the association patterns of diatom sub-communities with environmental parameters revealed 3 modules positively correlated with nutrient (nitrate, silica, phosphate) availability in both datasets. The V4 turquoise module, enriched in *Thalassiosira, Porosira* and *Chaetoceros* (~10% each), was positively correlated with nitrate, silica, phosphate and chlorophyll *a*. Conversely, both V4 modules

midnightblue – enriched in *Pseudo-nitzschia* (16%) *Actinocyclus* (14%) and *Corethron* (12%) – and purple – enriched in *Actinocyclus* (10%) and *Chaetoceros* (9%) – displayed a similar positive correlation with nitrate and phosphate while the latter was negatively correlated with iron and latitude. This potentially indicates that these diatom sub-communities are the most responsive to nutrient availability while being associated

**Fig. 6 | Abundance, diversity and distribution of the top 20 most abundant diatom genera in V4 and V9 datasets.** These genera accounted for 97% of assigned reads in the entire V4 dataset and 98% in the V9 dataset when pooling together all size fractions and depths. **a** Read taxonomic distribution. **b** Diversity and spatial distribution. Genus names are colour-coded according to diatom class. *a.* Abundance rank of each genus within each dataset (in black when ranking as top 20, otherwise in grey). *b.* Sum of the percentage of reads across all samples (log scale). *c.* Number of stations in which the genera are detected. *d.* Sum of the percentage of reads across oceans. Bars are colour-coded by the ocean. Comparison between microscopy and molecular methods for detecting the top 20 most abundant diatoms in the 20–180 µm size fraction. a. Abundance rank of each genus

within each dataset (in black when ranking as top 20, otherwise in grey). **b** Sum of the normalized reads (% for the genus among total diatoms) across all samples. Only samples with both microscopy and molecular methods were analysed (67 samples from the surface and DCM from 59 stations). Due to similar morphologies, we merged the counts for *Actinocyclus*, *Coscinodiscus,* and centric diatoms > 30 µm in length. The microscopy-based identification of *Fossulaphycus* (previously *Fossula*, and neither cultivated nor sequenced) was reclassified as *Fragilaria* due to similar morphologies and co-occurrence (microscopy detection of *Fossulaphycus* only at one Arctic station, where V4 detected a high abundance of *Fragilaria* reads).

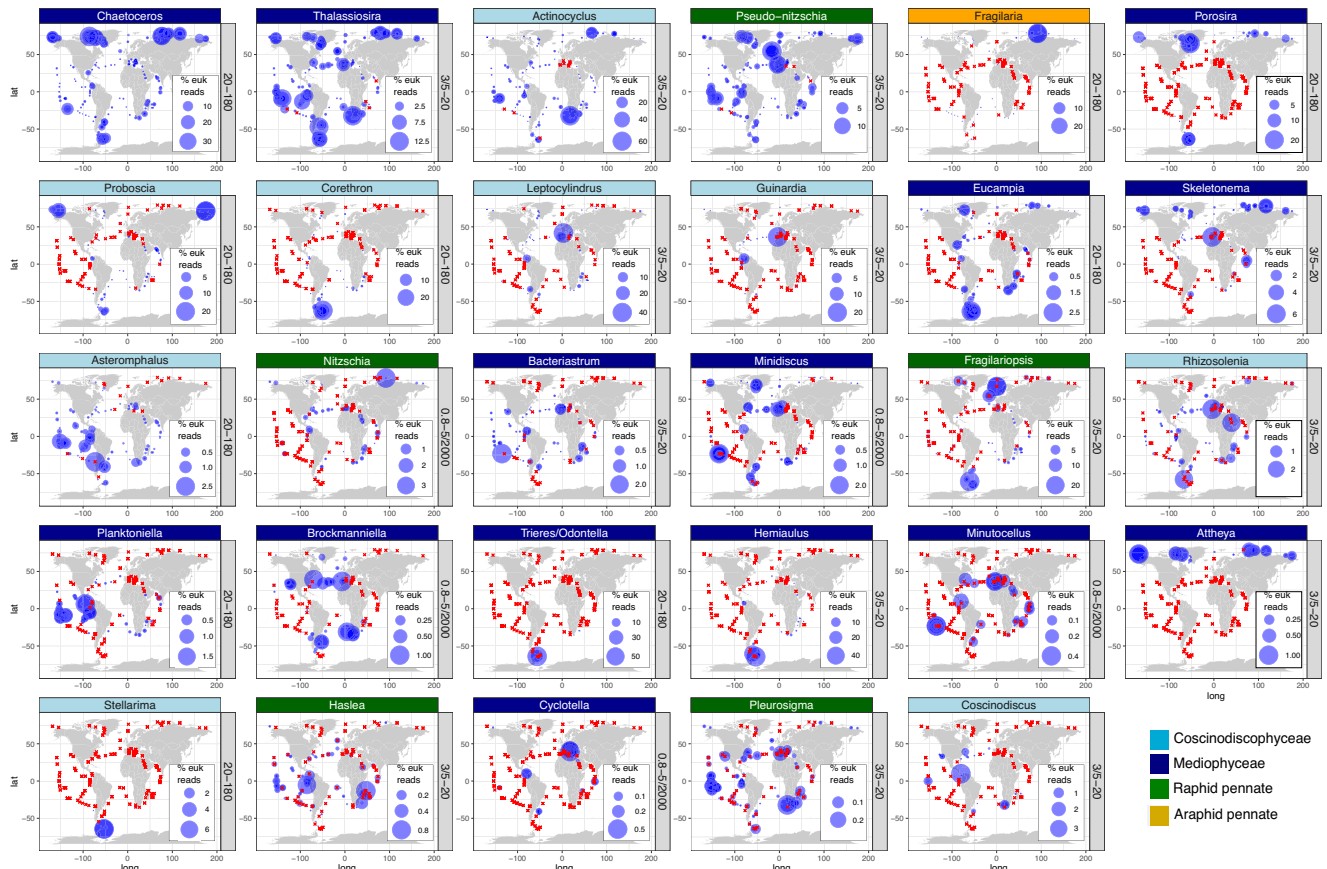

**Fig. 7 | Global distribution of the 20 most abundant genera in the V9 and V4 datasets.** All genera in the list of Fig. 6a are displayed. Bubble areas are scaled to the % reads for the genus among eukaryotic reads at each station location, whereas crosses indicate the absence of detection. The maps correspond to samples from the size fraction where the genus was most prevalent, indicated alongside each map. The maps include the circles for both V4 and V9 datasets, while separated maps by marker region are displayed in Supplementary Fig. S11. The maps, including all size fractions and genera, can be found in Supplementary Data 1. Maps were generated with the *borders()* function in *ggplot2*[89].

with different oceanic regions – either SO (midnight blue, turquoise) or SPO/NPO (purple).

On the other hand, all V9 pink, brown and turquoise modules, which all harboured species representative of SO basins and clustered together, were significantly correlated with the three nutrients but were enriched in a different set of species. While both V9 pink (SO/IO) and brown (SO/NAO) modules contained larger diatoms and showed a high proportion of *Chaetoceros* ASVs (49% and 13% of ASVs, respectively, ranked 2nd and 3rd), the latter also displayed a significant contribution of *Odontella* ASVs (7%) and was positively correlated with chlorophyll *a* and ammonium. In contrast, the V9 turquoise (SO/SPO) module was enriched in *Hemiaulus* and *Thalassiosira* ASVs (~10% and 7%). This, therefore, suggests the existence of 3 distinct SO-associated diatom sub-communities similarly responding to the

availability of major nutrients but differing by their size, taxonomic composition and response to other environmental factors.

Both V4 and V9 blue modules were enriched in *Chaetoceros* and *Thalassiosira* genera from the AO, corroborating their negative correlation with temperature, and showed positive correlations with dissolved inorganic carbon and chlorophyll *a*. The V9 blue module was also positively correlated with net primary productivity, as was the V4 royal blue module (~15% *Bacteriastrum*), illustrating the diatom taxa associated with the most productive regions. These could correspond to either species outcompeted in conditions of nutrient scarcity, and/or to taxa highly contributing to net primary productivity.

Finally, the V4 lightcyan module, which gathered very few ASVs (47) and notably from the genera *Chaetoceros* and *Guinardia*, was the

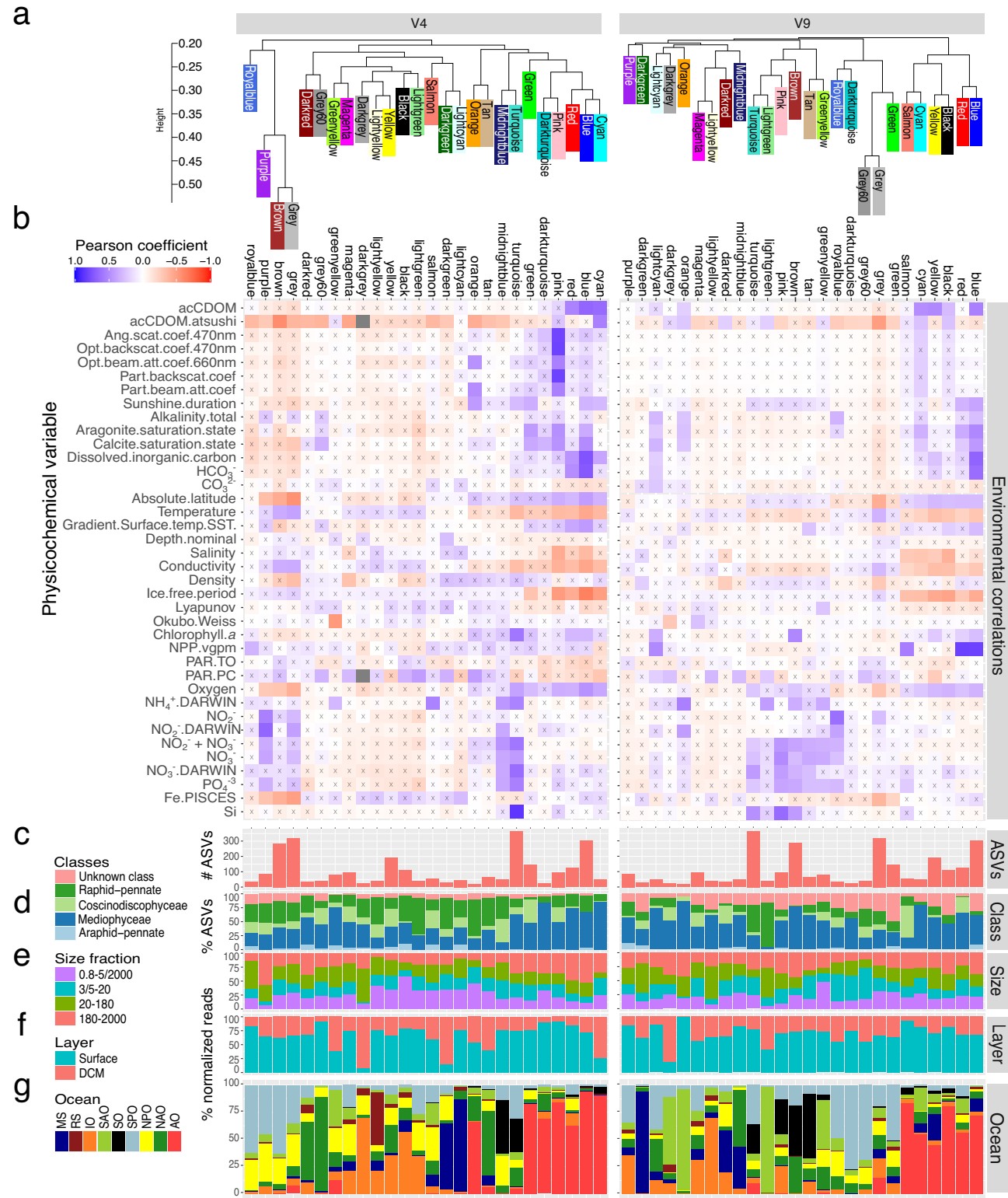

most associated with iron but showed no other significant pattern, which could either indicate taxa specialized in iron uptake or species with elevated iron quotas. Altogether, these analyses provide novel insights into factors regulating the assemblage of diatom communities and highlight modules harbouring species indicative of different environmental gradients and, therefore, showing different life strategies. These results could be considered for future studies exploring potential indicator species among plankton.

**Unassigned sequences**

Unassigned ASVs, which could not be classified at genus or species level, constituted 42.7% and 58.8% of the total diatom V4 and V9 ASVs, and 24% and 20.7% of total diatom V4 and V9 reads (Supplementary Fig. S13a). These include ASVs identified only at the phylum level (e.g., 'unknown' diatoms), which consist of 438 V4 and 1056 V9 ASVs. In addition, ASVs classified only at the class level included 1,694 V4 and 1,270 V9 ASVs. Notably, the number of unclassified ASVs from pennate

**Fig. 8 | Weighted gene correlation network analysis (WGCNA) of diatom communities for the V4 (left) and V9 datasets (right). a** Module clustering according to their co-occurrences. **b** Pearson correlation between module eigenvalues and physico-chemical parameters. Crosses indicate adjusted *p*-value > 0.05. **c** Total number of ASVs. **d** Percentage of ASVs for each diatom class. **e** Sum of the percentage of reads across size fractions. **f** Sum of the percentage of reads across water layers. **g** Sum of the percentage of reads across oceans. Abbreviations: acCDOM, absorption coefficient of coloured dissolved organic matter; Ang.scat.coef.470 nm, Angular scattering coefficient 470 nm; opt.backscat.coef.470 nm, Optical backscattering coefficient 470 nm; Opt.beam.att.coef.660 nm, Optical beam attenuation coefficient 660 nm; Part.backscat.coef, Backscattering coefficient of particles; Part.beam.att.coef, the Beam attenuation coefficient of particles; Gradient.Surface.Temp.SST, Sea surface temperature gradient; Lyapunov, maximum Lyapunov exponent; Okubo-Weiss, Okubo-Weiss parameter; NPP.vgpm, Net primary production from VGPM product; PAR, photosynthetic active radiation; NH4 + .DARWIN, modelled ammonium concentration; NO2-.DARWIN, modelled nitrite concentration; NO3-.DARWIN, modelled nitrate concentration; NH4 + , NO2- + NO3-, measured concentration of nitrate plus nitrite; NO3-, measured nitrate concentration; Fe.PISCES, Fe concentration from PISCES model.

diatoms exceeded those assigned to specific pennate genera/species (904 *vs* 666 ASVs for V4, and 634 *vs* 213 V9 ASVs). While planktonic pennate genera such as *Thalassionema, Fragilariopsis*, and *Pseudonitzschia* were abundant, pennate diatoms are overall more diverse and abundant in benthic environments[60]. The high number of unclassified pennate diatom ASVs in our dataset is consistent with the limited number of sediment studies, which are crucial to contribute to sequence diversity in reference datasets[56].

The proportion of unassigned reads was higher in the 3/5–20 µm size fraction and less prevalent in the larger size classes (20–180 µm and 180–2000 µm) (Supplementary Fig. S13b). This is an expected result because diatoms belonging to the largest size fractions are those more intensively studied and are, hence, better represented in genetic reference databases. In addition, there were variations across oceanic regions and sampling sites, with SPO, SAO, AO, and in particular SO, showing higher relative abundances of unassigned ASVs (Supplementary Fig. S13c), probably because the southern hemisphere and the poles are the least explored areas. While the V4 and V9 datasets showed similar trends for unassigned sequences overall, a few stations (TARA_085 and TARA_086 in the SO, TARA_188 in the AO, and TARA_006 in the MS) had a high abundance of unassigned V9 ASVs compared to V4 ASVs, while the opposite was found in TARA_158 in AO (Supplementary Fig. S13d). Some of these unknown ASVs are particularly abundant and should be prioritized in the future for further study.

## The biogeography of diatoms is consistent with their genetic and transcriptional features

Beyond taxonomic insights from 18S V4 and V9 sequences, complementary functional information can be derived from metatranscriptomics, which quantifies mRNA levels. Therefore, we mined polyA RNA sequencing data from the same plankton samples used for metabarcoding. We focused on the top 100 nuclear-encoded Pfam families with the highest transcript abundances, accounting for 56% of the total metatranscriptomic abundance for diatoms in the global ocean. Thus, this represents the cellular functions and processes in which natural diatom populations invest the bulk of their energy and resources, at least as viewed at the level of mRNA abundance (Fig. 9).

To reduce redundancy, Pfams with similar functions were merged. Specifically, 137 Pfams coding for different ribosomal subunits were grouped under "Ribosomal proteins", and 47 Pfams coding for various ubiquitin domains were combined into "Ubiquitin domains". As a result of this merging process, these two functional categories rank higher, representing 9.2% and 4.2% of total diatom transcript abundance, respectively. However, at the individual level, the most abundant Pfam family was the light-harvesting chlorophyll a/b-binding complex (LHC) proteins (PF00504). LHCs, proposed to be the most abundant membrane proteins on Earth, accounted for almost 3% of diatom metatranscriptomic abundance.

Additional photosynthetic families within our top 100 Pfams include ATP synthase subunit C (PF00137), which plays a role in carbon-concentrating mechanisms[67], and flavodoxin (PF00258), contributing 0.24% and 0.20% of diatom metatranscriptomic abundance. Beyond photosynthesis, the maintenance of the frustule also appeared

to require a high transcriptional effort, reflected by the observation that the silicon transporters (PF03842)[68] were the 40th most transcriptionally abundant family (0.36% of diatom metatranscriptomic abundance), and are mainly found in diatoms and dictyochophytes (another silicifying group).

Among the top 100 Pfams, only one domain of unknown function (DUF) was present: DUF285 (PF03382). This enigmatic tandem repeat protein is distantly related to leucine-rich repeats[69] (Fig. 10a) and has multiple gene copies in some bacteria, giant viruses and in eukaryotic phytoplankton (Supplementary Fig. S14). Genes and transcripts coding for DUF285 are notably abundant at high latitudes (Fig. 10b, c), and this trend extends to phytoplankton beyond diatoms (Supplementary Fig. S15). Interestingly, DUF285 exhibits a solenoid conformation (Fig. 10a) similar to proteins with antifreeze activity[70].

Other Pfam families displayed specific transcript abundance patterns based on ocean regions, such 'Cold-shock' DNA-binding domains (PF00313) being predominant in AO and SO, and the opposite trends for Heat Shock Factor-type DNA binding domains (PF00447) and Heat Shock Proteins 20, 70 and 90 (HSP20, HSP70 and HSP90; PF00011, PF00012 and PF00183, respectively) (Fig. 9 and Supplementary Fig. S15). Other Pfams more abundant in temperate waters include many of the diatom prevalent families such as dioxygenase (PF00775), retinal pigment epithelial membrane protein (PF03055), subtilase family (PF00082), YHYH protein (PF14240), Zinc finger C2H2 type (PF00096), and zinc-binding domain (PF12907).

Given the dominance of LHCs, we conducted a detailed analysis of this family. In addition to the canonical members involved in light-harvesting (LHCf, LHCq (also known as LHCy), and LHCr), this superfamily includes the diatom-specific LHCx and LHCz subfamilies, engaged in photoprotection[71] (Fig. 10d). After annotating over 52,000 diatom LHC sequences into these subfamilies, distinct expression patterns emerged. The ratio of transcripts for light-harvesting subfamilies to those for photoprotection subfamilies was higher in larger size fractions (Fig. 10e–g). This increase likely serves as a compensatory response to counteract intracellular light attenuation: as cell volume increases, the path length of light within the cell also increases, reducing the light intensity per unit volume. We also observed spatial distinctions in expression related to depth and latitude. Among the pool of LHC transcripts, photoprotective subfamilies were prevalent at the surface, whereas light-harvesting subfamilies were predominant in the DCM (Fig. 10f). In addition, photoprotection subfamilies were slightly more prevalent at higher latitudes, which are areas characterized by photo-physiological stresses such as low temperatures and elongated photoperiods (long days in the summer, and long nights in the winter) (Fig. 10e).

To infer adaptation patterns (e.g., gene copy number variations) in different diatom species, we explored the metagenome-assembled genomes (MAGs) recently reconstructed from the same DNA samples used for amplifying the 18S rRNA gene amplicons[72]. Due to assembly and binning challenges, these MAGs lack the 18S rRNA gene, so their taxonomy was inferred using protein-coding marker genes[72]. A total of 52 MAGs were assigned to diatoms, with variable ranges of completeness (3-87% BUSCO score[73]). Focusing on the 18 MAGs with over

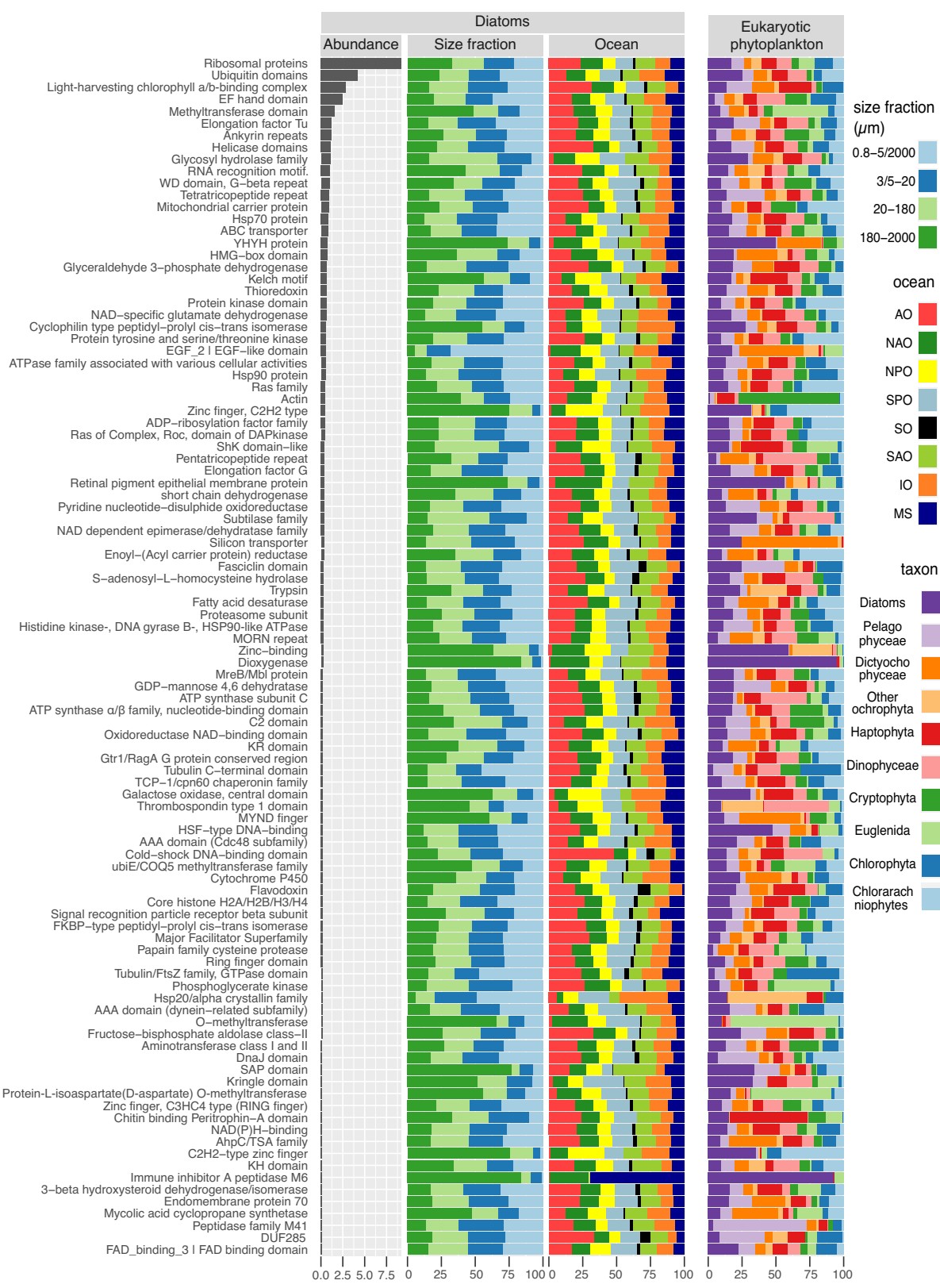

70% BUSCO completeness, we observed a slight tendency towards higher copy numbers of genes encoding LHCx, LHCz and DUF285 in MAGs common in high latitudes (Fig. 10h). While additional data—more MAGs with higher completeness—are needed to confirm these trends, the observed biogeography of the MAGs seem to align with these genetic characteristics, further underlining the intricate relationship between genetic adaptation and environmental factors in diatom populations.

## Discussion

This study utilizes the *Tara* Oceans datasets to provide an extensive analysis of diatom diversity and distributions across the world's ocean,

**Fig. 9 | Top 100 most expressed diatom Pfam families in the global ocean.**
(Column a) Abundance ordered by rank. (Column b) Abundance across size fractions. (Column c) Abundance among ocean regions (AO, Arctic Ocean; NAO, North Atlantic Ocean; NPO, North Pacific Ocean; SPO, South Pacific Ocean; SO, Southern Ocean; SAO, South Atlantic Ocean; IO, Indian Ocean; MS, Mediterranean Sea). (Column d) Abundance in diatoms vs other eukaryotic phytoplankton. For each sample, the abundance of a given diatom Pfam was calculated as the percentage of total metatranscriptomic reads assigned to diatoms in that sample. The same calculation was applied to other phytoplankton for column d. Note that we were unable to exclude non-photosynthetic species from dinoflagellates due to the limited number of reference gene sequences available for this group. To reduce redundancy, Pfams for similar functions were merged (e.g., 137 Pfams coding for different diatom ribosomal subunits were merged into "Ribosomal proteins", and 47 Pfams coding for ubiquitin domains were merged into "Ubiquitin domains").

enhanced by functional profiling. We employed parallel metabarcoding analyses of the V9 and V4 hypervariable regions, offering a wide range of taxonomic resolution ideal for genus-level diversity research. Despite differences in taxonomic resolution and reference databases, our results revealed significant similarities between V4 and V9 molecular patterns, with 15 genera common among the top 20 in both datasets. Additionally, these molecular methods were corroborated with light microscopy data. Our findings reinforce the role of diatoms as a key photosynthetic group in marine ecosystems, highlighting prevalent genera such as *Chaetoceros*, *Thalassiosira*, *Actinocyclus*, and *Pseudo-nitzschia*. Correlation analyses indicated that nutrient limitations and biotic interactions significantly shape diatom relative abundance, particularly through competition with radiolarians for silicon and niche differentiation with picocyanobacteria over nitrogen forms. We defined 25 distinct communities of co-occurring diatom lineages with varying environmental preferences, illustrating diverse ecological adaptations. The AO stands out as a hotspot of diatom abundance and diversity, with 6 of the diatom communities exclusive to it. Our results provide insights into diatom life strategies from a wide range of ecosystems and illustrate the ecological convergence of diatom communities from distinct oceanic regions regarding the availability of major nutrients. We envision that these results will help to identify indicator species for ocean monitoring, especially since many diatom lineages were predicted to be vulnerable to climate change scenarios[74].

In addition to taxonomic insights from metabarcoding, we obtained functional information from metatranscriptomics by analysing changes in mRNA abundance in environmental samples. Our data indicate distinct gene expression patterns correlating with biogeographical distribution. For instance, diatoms in higher latitudes exhibited increased transcript levels coding for 'Cold-shock' DNA-binding proteins as well as for the enigmatic DUF285, suggesting a link between metabarcoding patterns and gene expression profiles. Globally, the most expressed gene family was LHC, with varying expression patterns among subfamilies related to light harvesting versus photoprotection, reflecting changes in depth, latitude and cell size. To understand adaptation strategies, such as gene copy number variations among diatom species, we analysed MAGs derived from the same DNA samples used for 18S rRNA gene amplification. The biogeographical distribution of diatom MAGs showed some alignment with their gene copy numbers related to photoacclimation, but more MAGs with higher completeness are needed to confirm these trends.

In conclusion, our study offers a thorough pole-to-pole examination of marine diatom populations, encompassing the entire plankton size spectrum. This research can reveal important insights into the impact of global changes and human-induced disturbances on marine ecosystems.

## Methods
### *Tara* Oceans sampling
Tara Oceans expeditions between 2009 and 2013 performed a worldwide sampling of plankton in the upper layers of the ocean, including surface (SRF; 5 m depth) and deep chlorophyll maximum (DCM; 17–188 m). To capture the whole size spectrum of plankton, a combination of filter membranes with different pore sizes (size-fractionation) was used[36]. Four major eukaryotic-enriched size fractions

were collected: pico-nanoplankton (0.8–5 μm or 0.8–2000 μm; named here 0.8–5/2000 μm), nanoplankton (5–20 μm or 3–20 μm; named here 3/5–20 μm), microplankton (20–180 μm), and mesoplankton (180–2000 μm)[3,36]. Given the inverse relationship between plankton size and abundance[36], the seawater volumes filtered was increased for larger size fractions (from $10^2$ to $10^5$ L; see Table 1 and Fig. 5 in Pesant et al.[36]). These plankton samples were leveraged to generate different molecular and optical datasets analysed in the current work (Supplementary Fig. S2). Specific details about them and their analyses are described below.

### Metabarcoding datasets
Nucleic acid extraction methods were applied to obtain DNA from the different size-fractionated samples. In order to target eukaryotic plankton diversity and relative abundance, a metabarcoding approach was performed using the 18S rRNA gene as a molecular marker. Detailed information about DNA extraction, PCR amplification and Illumina sequencing of metabarcodes are described by refs. 75,76. Briefly, DNA samples were amplified by PCR targeting the hypervariable region V9 (130 ± 4 base pairs length; primer pair: 1389 F 5′-TTGTACACACCGCCC-3′ and 1510 R 5′-CCTTCYGCAGGTTCACCT AC-3′)[77] or V4 (385 ± 4 base pairs length; primer pair TAReuk454FWD1 5′-CCAGCASCYGCGGTAATTCC-3′ and TAReukREV3 5′-ACTTTCGTT CTTGATYRA-3′)[78] of the 18S rRNA marker gene followed by the Illumina sequencing of the amplicons[75]. The details of PCR mixes, thermocycling and sequencing conditions are provided in Alberti et al.[76].

The Resulting paired-end reads were mixed-oriented meaning that both R1 and R2 files are composed by a mix of forward and reverse reads. Paired-end reads were trimmed to remove PCR primer sequences using Cutadapt v2.7[79] and dispatched into four files, 2 files for the classical orientation (forward reads in R1 and reverse reads in R2) and 2 others for the other orientation (reverse reads in R1 and forward reads in R2). Paired-end reads without both primers were filtered out using the option –discard-untrimmed. Forward and reverse reads were trimmed at position 80 for V9 and at position 215 for V4 and reads with ambiguous nucleotides or with a maximum number of expected errors (maxEE) superior to 2 were filtered out using the function *filterAndTrim*() from the R package *dada2*[37]. For each run and read orientation, error rates were defined using the function *learnErrors*() and denoised using the *dada*() function with pool = TRUE before being merged using *mergePairs*() with default parameters. Mixed orientated reads from the same sample and sequencing replicates were summed together. Remaining chimeras were removed using the function *removeBimeraDenovo*(). Scripts producing the ASV tables are publicly available here: https://gitlab.univ-nantes.fr/combi-ls2n/taradada. ASVs were taxonomically assigned using IDTAXA (50% confidence threshold, no sequence similarity threshold)[38] with the PR2 database version 4.14[39]. The scripts for the taxonomic assignment are publicly available here: https://gitlab.sb-roscoff.fr/nhenry/abims-metabarcoding-taxonomic-assignment/-/tree/v1.0.1. To facilitate a comparison between diatoms and other photosynthetic groups, we annotated the ASVs as phytoplankton—including photosynthetic dinoflagellates and chrysophytes—whenever their taxonomic resolution was adequate for alignment with known phytoplanktonic lineages.

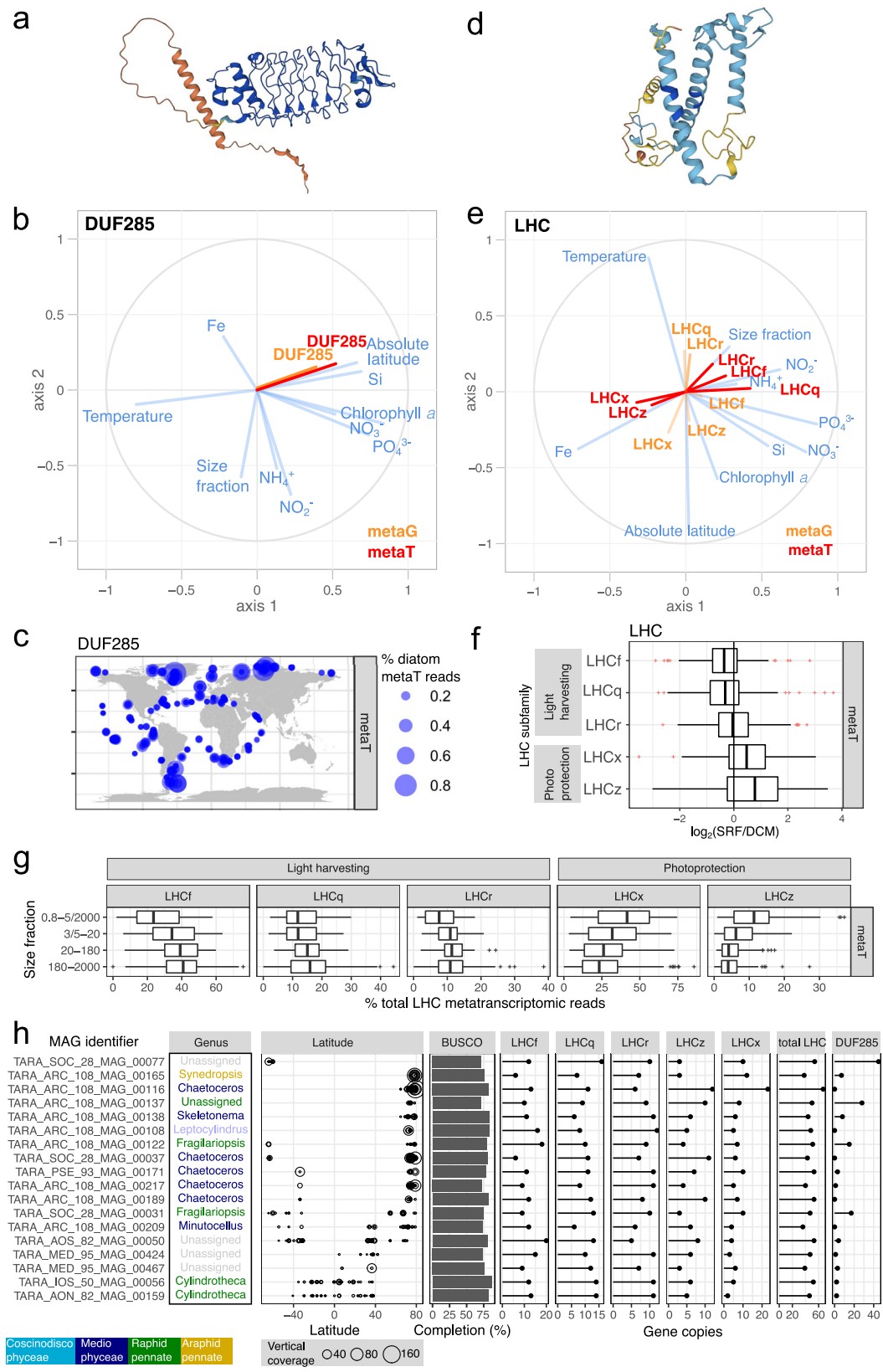

This classification was informed by a literature review and expert insights compiled in the trait reference database accessible at https://doi.org/10.5281/zenodo.3768950. To reduce the influence of PCR and sequencing errors not detected by the dada2 algorithm, only ASVs seen in at least two different samples with at least three copies were retained.

**Metatranscriptomic and metagenomic datasets**

We mined the *Tara* Oceans catalogue of eukaryotic expressed genes (Marine Atlas of *Tara* Oceans Unigenes; MATOU[80,81]). Version 1.5 of MATOU, released alongside this publication (https://www.genoscope.cns.fr/tara/#MATOU-1.5), includes as an important update the incorporation of samples from the Arctic Ocean.

**Fig. 10 | Gene and transcript patterns for the Pfam families DUF285 and LHC among diatom populations. a–c** DUF285 (PF03382). **a** Structural prediction (Uniprot id B7FV04). **b** PLS analysis for metagenomic (orange) and metatranscriptomic (red) abundances. **c** Metatranscriptomic abundance biogeography. **d–g** LHC (PF00504). **d** Structural prediction (Uniprot id B7G6Y1). **e** PLS analysis for metagenomic (orange) and metatranscriptomic (red) abundances. **f** Depth partition between surface (SRF) and deep chlorophyll maximum (DCM). **g** Partition across size fractions. **h** Gene copy patterns for LHC and DUF285 in metagenomic-assembled genomes (MAGs) assigned to diatoms. The columns correspond to: (i) MAG identifier. (ii) Class and genus. (iii) Latitudinal distribution. Bubbles area is proportional to vertical read coverage. iv) Completeness based on BUSCO[74]. Only MAGs with > 70% completeness are displayed. (v) Gene copies for LHC subfamilies and for DUF285. Boxplots illustrate the distribution of the dataset, with the box representing the 25–75% interquartile range and the central line indicating the median (50% quantile). Whiskers extend to data points within 1.5 times the interquartile range. The metatranscriptomic dataset comprises 121 samples for the 0.8–5 μm or 0.8–2000 μm size fractions, 110 for the 3–20 μm or 5–20 μm fractions, 116 for the 20–180 μm fraction, and 121 for the 180–2000 μm fraction. Maps were generated with the *borders()* function in *ggplot2*[89].

Briefly, the catalogue was built in the following steps. Nucleic acid extraction methods were applied to obtain eukaryotic polyA RNA from the different size-fractionated samples and subsequent Illumina sequencing was carried out[76,80]. The corresponding metatranscriptomic reads were assembled and clustered at 95% identity[80,81]. The metatranscriptomic reads were mapped onto the sequence catalogue to provide the abundance of each transcript sequence in every sample[80]. Similarly, metagenomic reads from the same seawater samples were mapped onto the sequence catalogue to ascertain the abundance of each gene in every sample[80]. The taxonomic classification of the unigene catalogue was carried out by sequence similarity against a custom reference database comprising METdb92[82], UniRef90 (version 20190810), and single-cell amplified genomes (SAGs) from *Tara* Oceans[83]. Functional assignation was performed using the Pfam database[80].

For our analysis, we focused on sequences assigned to diatoms and other eukaryotic phytoplankton groups for comparative purposes. It is important to note that we could not exclude heterotrophic dinoflagellate species due to the limited number of reference sequences available for this group. We then retrieved the metatranscriptomic read abundances (rpkm values) of the selected sequences and normalized them to the total read abundance for transcripts of the corresponding phytoplankton taxon in each sample. Additionally, for specific cases of interest such as LHC, DUF285, and HSP90, we also retrieved the metagenomic read abundances and performed the equivalent abundance normalization.

## Metagenome-assembled genomes (MAGs) dataset

MAGs were reconstructed and manually curated from *Tara* Oceans metagenomic reads by Delmont et al.[72] (available at https://www.genoscope.cns.fr/tara/#SMAGs). The MAGs received a geographical assignment based on their read recruitment in the *Tara* Oceans sampling stations after mapping each MAG onto the *Tara* Oceans metagenomic dataset[72]. The estimation of genome completion was performed by retrieving the Benchmarking Universal Single-Copy Orthologs (BUSCO)[73]. The taxonomic annotation was carried out by phylogenetic analyses of eukaryotic core gene markers[72]. For the current analysis, we selected the 18 MAGs assigned to diatoms and with over 70% BUSCO completeness, and we determined their gene copy number for LHCs and DUF285 as described below.

## Optical microscopy datasets

For light microscopy, three ml of each sample (from 20-180 μm size fractions) were placed in an Utermöhl chamber. Cells falling in 2 or 4 transects of the chamber were identified and enumerated using an inverted light microscope (Carl Zeiss Axiophot200) at 400x magnification. To be compared with the molecular data, the optical microscopy counts of diatoms were expressed as percentages of the genera (or of each genus) over the total diatom community (%).

## WGCNA of diatom metabarcoding data

We used the R package WGCNA (Weighted Gene Co-Expression Network Analysis)[65] to infer the ASVs co-occurence networks for both V4 and V9 metabarcoding data. To remove redundancy, for every station we pooled the reads for the four size fractions into a single aggregated sample. We discarded the samples without all the four size fractions. A Pearson correlation matrix was computed using the ASVs abundances. We raised the Pearson correlation to a power of 12 and 20, for V4 and V9 respectively, to compute the adjacency matrices and we used these matrices to generate a topological overlap measure (TOM). A hierarchical clustering was finally performed on the TOM matrices to identify 25 modules and a set of unclustered ASVs (grey module) (Fig. 8). The module eigengenes were also computed for each module identified (grey module included) and then used to generate correlations (and associated *p*-values) between each module and 48 environmental variables retrieved from https://doi.org/10.1594/PANGAEA.875582 (Figure S16) (see also below).

## Analysis of LHC and DUF285 in metagenomes and metatranscriptomes

We analysed the gene and transcript abundances of LHCs and DUF285 according to environmental variables by mining the metatranscriptomes and metagenomes generated by *Tara* Oceans. In particular, we searched the MATOU-v1.5 https://www.genoscope.cns.fr/tara/#MATOU-1.5 catalogue for sequences coding for LHCs (PF00504) and DUF285 (PF03382) by running HMMer (version 3.2.1 with the gathering threshold option; http://hmmer.org/) over the translated sequences. We retained only sequences assigned to diatoms, and retrieved the corresponding metagenomic and metatranscriptomic read abundances (rpkm values) and normalized them to the total read abundance for diatom transcripts in each metatranscriptomic sample. We did a similar HMMer search among the gene catalogue from diatom MAGs.

For the functional classification of LHCs into the main subfamilies (LHCf, LHCq, LHCr, LHCx and LHCz), we performed a phylogenetic placement of the translated sequences on the reference phylogeny described in Kumazawa et al.[71]. Protein sequences were aligned with mafft version 6 using the G-INS-I strategy[84]. The output alignment was trimmed in both N- and C-terminal regions to maintain the reference alignment limits. The alignment was also processed using trimAl v1.495 with the gap threshold option -gt 0.1 to keep columns where at most 10% of sequences contain a gap[85]. The phylogenetic inference was made using approximate maximum likelihood with FastTree[86]. The sequences were classified according to their grouping in monophyletic branches with statistical support of >0.7 with reference sequences of the same functional group.

## Plotting and statistical analysis

Analyses were carried out in R language (http://www.r-project.org/). Exponentiated Shannon Diversity Index (expH), a diversity index that accounts for both species richness and evenness was calculated using R library *vegan*[87]. Estimates of total ASV richness were done with the *specpool()* function from the *vegan* library. Graphical analyses were carried out using R library *ggplot2*[88], and treemaps were generated with *treemap*[89]. Maps were generated with the *borders()* function in *ggplot2*[88] and *geom_point()* function for bubbles or

*scatterpie* package[90] for pie charts. Loess smooth plots with 95% confidence windows were plotted with the *geom_smooth*() function in *ggplot2*. Spearman's rho correlation coefficients and *p*-values were calculated using the *cor.test()* function of the *stats* package. Metric multidimensional scaling (NMDS) analysis to visualize Bray–Curtis distances was carried out with the *metaMDS()* command in the R package *vegan*[91], and the influence of environmental variables on sample ordination was evaluated with the function *envfit*() in the same R package, whereas the pie charts were plotted with *scatterpie* package. Partial least square analyses were implemented with the R package *plsdepot*[92] to explore the correlations between diatom relative abundance and Shannon index with the physicochemical context (range-transformed median values of nutrient concentrations, chlorophyll a, absolute latitude, and temperature). A second set of partial least square analyses were performed to explore the correlation of diatom relative abundance to the relative abundance of other silicifiers, and the abundances of picocyanobacteria, copepods and total Rhizaria (see below).

### Physico-chemical and biotic parameters

Physicochemical parameters were retrieved from https://doi.org/10.1594/PANGAEA.875582, and a full description of the performed measurements is described in Pesant et al.[36]. Measurements of temperature, conductivity, salinity, depth, pressure, and oxygen were carried out with a vertical profile sampling system (CTD-rosette) and Niskin bottles. Chlorophyll concentration, a proxy for total phytoplankton biomass, were measured using high-performance liquid chromatography. Dissolved nutrients ($NO_3^-$, $PO_4^{3-}$, Si) were analysed according to previous methods[93,94]. We complemented the in situ measurements with silica, nitrate, nitrite, ammonium, and iron levels derived from the ECCO2-DARWIN ocean model[95].

For biological predictors, we included the abundance of picocyanobacteria (*Synechococcus* and *Prochlorococcus*), copepods (grazers), and silicifiers. Copepod counts were collected with the WP2 net (mesh size of 200 μm) towed vertically from 500 m to the surface (http://ecotaxa.obs-vlfr.fr/prj/377, http://ecotaxa.obs-vlfr.fr/prj/378). Picocyanobacteria counts were determined by flow cytometry[12] (https://data.mendeley.com/datasets/p9r9wttjkm/2). The relative abundances of silicifiers were based on the percentage of reads among V4 and V9 barcode datasets as described above.

### Reporting summary

Further information on research design is available in the Nature Portfolio Reporting Summary linked to this article.

## Data availability

The authors declare that all the data supporting the findings of this study are publicly available in the following repositories and in the supplementary information files of this paper. The contextual data are available in Pangaea (www.pangaea.de) with the identifier https://doi.org/10.1594/PANGAEA.875582, and a simplified version is available in https://zenodo.org/records/7229815. rDNA 18S metabarcoding data are deposited at the European Nucleotide Archive under accession numbers PRJEB6610 and PRJEB9737, and metagenomic and metatranscriptomic data under PRJEB402, PRJEB9691, PRJEB9738 and PRJEB9739. The ASV table and the taxonomic annotation for the V4 marker are available in https://zenodo.org/records/13881376 and for the V9 marker in https://zenodo.org/records/13881418. The files for the Marine Atlas of *Tara* Oceans Unigenes version 1.5 (MATOU-v1.5) are available in https://www.genoscope.cns.fr/tara/#MATOU-1.5, including the FASTA sequences, taxonomic and functional annotation tables, as well as metagenomic and metatranscriptomic abundance tables. The manually curated MAGs by Delmont et al.[72] are available at https://www.genoscope.cns.fr/tara/#SMAGs. Flow cytometry data are

available at https://data.mendeley.com/datasets/p9r9wttjkm/2. Source data are provided in this paper.

## Code availability

Scripts producing the ASV tables are publicly available here: https://gitlab.univ-nantes.fr/combi-ls2n/taradada. The scripts for the ASV taxonomic assignment are publicly available here: https://gitlab.sb-roscoff.fr/nhenry/abims-metabarcoding-taxonomic-assignment/-/tree/v1.0.1. Scripts for producing the figures in this paper are in https://github.com/JJPierellaKarlusich/Diatom_patters/tree/main (Pierella Karlusich, JJ, Patterns and drivers of diatom diversity and abundance in the global ocean, Zenodo, https://doi.org/10.5281/zenodo.14890007, 2025).

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

## Acknowledgements

We would like to thank all colleagues from the *Tara* Oceans consortium as well as the Tara Ocean Foundation for their inspirational vision. We are grateful to Shruti Malviya and Federico Ibarbalz for providing initial details about the metabarcoding data. This work was granted access to the HPC resources of IDRIS, Jean Zay supercomputer, under the allocation 2020–DARI A0080111497 made by GENCI, and has been supported by the FFEM - French Facility for Global Environment, French Government 'Investissements d'Avenir' programmes OCEANOMICS (ANR-11-BTBR-0008), FRANCE GENOMIQUE (ANR-10-INBS-09-08), MEMO LIFE (ANR-10-LABX-54), and PSL Research University (ANR-11-IDEX-0001-02). C.B. acknowledges funding from the European Research Council (ERC) under the European Union's Horizon 2020 research and innovation programme (Diatomic; grant agreement No. 835067), the Horizon Europe projects Marco-Bolo (grant agreement no. 101082021) and Blue-Remediomics (grant agreement no. 101082304), Agence Nationale de la Recherche "Phytomet" (ANR-16-CE01-0008) and BrownCut (ANR-19-CE20-0020), and the BNP Paribas Foundation Climate & Biodiversity Initiative. J.J.P.K. acknowledges postdoctoral funding from the Fonds Français pour l'Environnement Mondial and by Moore–Simons Project on the Origin of the Eukaryotic Cell, Simons Foundation (735929LPI). This article is contribution number 156 of *Tara* Oceans.

## Author contributions

C.B. and J.J.P.K. designed the study and supervised the project. T.O.C. organized sample collection and analysis. J.J.P.K. and C.B. wrote the paper with substantial input from L.Z., A.Z., and K.C. and all co-authors. L.Z. and A.Z. helped with new ideas and result interpretation. A.Z. and E.S. assisted with the optical microscopy data and interpretation. S.C. and Erwan Delage built the ASV tables, and NH performed the taxonomic annotation. K.C. ran initial exploratory molecular analyses. G.B. ran the WGCNA calculations, J.J.P.K. plotted the results, and C.N. and J.J.P.K. performed manual analysis and provided interpretations. F.R.J.V. and Etienne Dvorak ran an initial exploratory analysis of metatranscriptomics, and S.O. helped with result interpretation. J.J.P.K. performed the formal analysis and visualization.

## Competing interests

The authors declare no competing interests.

## Additional information

## Tara Oceans Coordinators

Silvia G. Acinas[9], Marcel Babin[2,10], Peer Bork[11,12,13], Emmanuel Boss[14], Chris Bowler[1,2], Guy Cochrane[15], Colomban de Vargas[2,16], Gabriel Gorsky[17], Nigel Grimsley[18,19], Lionel Guidi[2,17,20], Daniele Iudicone[6], Olivier Jaillon[2,21,22,23], Stefanie Kandels[24], Lee Karp-Boss[14], Eric Karsenti[1,25], Fabrice Not[26], Hiroyuki Ogata[27], Stéphane Pesant[28,29], Nicole Poulton[30], Christian Sardet[17], Sabrina Speich[2,31,32], Lars Stemmann[2,17], Matthew B. Sullivan[33], Shinichi Sunagawa[34] & Patrick Wincker[2,21,22,23]

[9]Department of Marine Biology and Oceanography, Institute of Marine Sciences (ICM)-CSIC, Pg. Marítim de la Barceloneta 37-49, Barcelona, Spain. [10]Département de biologie, Québec Océan and Takuvik Joint International Laboratory (UMI3376), Université Laval (Canada) - CNRS (France), Université Laval, Québec, QC, Canada. [11]Structural and Computational Biology, European Molecular Biology Laboratory, Meyerhofstrasse 1, Heidelberg, Germany. [12]Max-Delbrück-Centre for Molecular Medicine, Berlin, Germany. [13]Department of Bioinformatics, Biocenter, University of Würzburg, Würzburg, Germany. [14]School of Marine Sciences, University of Maine, Orono, ME, USA. [15]European Molecular Biology Laboratory, European Bioinformatics Institute (EMBL-EBI), Welcome Trust Genome Campus, Hinxton, Cambridge, UK. [16]Sorbonne Université, CNRS, Station Biologique de Roscoff, AD2M ECOMAP, Roscoff, France. [17]Sorbonne Université, CNRS, Laboratoire d'Océanographie de Villefanche, LOV, Villefranche-sur-mer, France. [18]CNRS UMR 7232, Biologie Intégrative des Organismes Marins, Avenue du Fontaulé, Banyuls-sur-Mer, France. [19]Sorbonne Universités Paris 06, OOB UPMC, Avenue du Fontaulé, Banyuls-sur-Mer, France. [20]Department of Oceanography, University of Hawaii, Honolulu, Hawaii 96822, USA. [21]Génomique Métabolique, Genoscope, Institut François Jacob, CEA, CNRS, Univ Evry, Université Paris-Saclay, Evry, France. [22]CNRS, UMR 8030, 2 rue Gaston Crémieux, Evry, France. [23]Université d'Evry, UMR 8030, CP5706, Evry, France. [24]EMBL Heidelberg Director's office, Meyerhofstrasse 1, 69117 Heidelberg, Germany. [25]Directors' Research European Molecular Biology Laboratory Meyerhofstr, Heidelberg, Germany. [26]Sorbonne Université, CNRS - UMR7144 - Ecology of Marine Plankton Group, Station Biologique de Roscoff, Place Georges Teissier, Roscoff, France. [27]Institute for Chemical Research, Kyoto University, Gokasho, Uji, Kyoto, Japan. [28]PANGAEA, Data Publisher for Earth and Environmental Science, University of Bremen, Bremen, Germany. [29]MARUM, Center for Marine Environmental Sciences, University of Bremen, Bremen, Germany. [30]Bigelow Laboratory for Ocean Sciences, East Boothbay, ME, USA. [31]Department of Geosciences, Laboratoire de Météorologie Dynamique (LMD), Ecole Normale Supérieure, Paris, Cedex 05, France. [32]Ocean Physics Laboratory, University of Western Brittany, 6 avenue Victor-Le-Gorgeu, BP 809, Brest, France. [33]Departments of Microbiology and Civil, Environmental and Geodetic Engineering, The Ohio State University, Columbus, OH, USA. [34]Department of Biology, Institute of Microbiology and Swiss Institute of Bioinformatics, ETH Zurich, Vladimir-Prelog-Weg 4, 8093 Zurich, Switzerland.

