## [Transparent Peer Review file · Nature Communications]

Patterns and drivers of diatom diversity and abundance in the global ocean

Corresponding Author: Dr Juan Pierella Karlusich

Version 0:

Reviewer comments:

Reviewer #1

(Remarks to the Author)

“Patterns and drivers of diatom diversity and abundance in the global ocean” by Karlusich et al. is a follow up to Malviya et al. 2016 (PNAS) in which diatom diversity across the global ocean was first presented. This current study aims to address underlying factors driving the abundance and functional diversity of marine diatoms by leveraging a more exhaustive global Tara Oceans data set (double the number of sites, includes a polar region, which was missed in the original analysis). They dig further into taxonomic diversity through amplicon sequences as well as characterize functional diversity through metatranscriptomics. The comparison of V4 and V9 18S rRNA amplicon sequencing results throughout the manuscripts is especially valuable from a methodological standpoint, as there is little consistency in the field in terms of which amplicon region is being applied to survey phytoplankton communities. These results are convincing that both amplicon regions yield similar results for the diatoms.

This study is able to achieve high level insights into diatom ecology and evolution that is only possible through the compilation and analysis of global data, and the results are noteworthy. The abundance and diversity correlations with other groups is especially interesting, including the negative relationship with cyanobacteria and radiolarians, and positive relationships with copepods. The biogeographical trends among lineages of diatoms is also insightful, as it provides information their distinct evolutionary trajectories (e.g., araphid pennates in sea ice).

I have several comments regarding normalization approaches and the MAG analysis that should be addressed by the authors. However I don't think this invalidates the study, and I recommend this manuscript for publication following revisions. Detailed comments are provided below.

I am wondering about normalization strategies within the V9 Tara datasets, and whether differences in community biomass should be accounted for. For example in Fig. 7, if the objective is to determine where these 20 taxa are relatively abundant and where they dominate the diatom community, wouldn't it make more sense to look at their read counts relative to total diatom reads? Diatoms are generally abundant in the Southern Ocean, so this might be why many of them appear elevated in this location when normalized to % eukaryotic reads. A few sentences about how the normalization strategy was chosen and any potential caveats would be helpful for readers to interpret biological trends.

The reported increase in LHC read copy numbers at high latitude is not obvious from Fig. 10h. I recommend either showing statistical support, or removing this metagenomic component from the text. It's highlighted in the abstract and conclusion so it is a strong selling point at the moment that I am not sure is supported by the analysis.

Detailed comments

Line 142: “Second most occurring group” – I don't see what is meant by this or which figure shows this

Line 152 and again in conclusion on line 541: In what way could diatoms be used as a bioindicator of ocean health for monitoring purposes. Is a very low relative abundance of diatoms an “unhealthy” ecosystem? More explanation is needed.

Line 160: This study has the light microscopy data to comment on whether their high relative abundance in 18S amplicons is likely related to having a high 18S copy number (and large biovolume). Are they also relatively abundant in image data?

Line 176: Could the similar number of ASVs also be related to V9 not being able to differentiate among strains (multiple genera/species share the entire region), while the longer but less deeply sequenced V4 can?

Line 213: These correlation results are really interesting, especially the concepts around competition for N with cyanobacteria and for Si with radiolarians. Can PLS correlation metrics be included in the text to quantify the strength of these relationships? Even just knowing which are very strong vs. moderate would be helpful.

Line 235: What about filtered biomass aggregating on filters and resulting in smaller porosity? This could be how small cells end up on large filters.

Line 438: "Pennate diatoms.. only occasionally found in the plankton" – this doesn't make sense given Pseudo-nitzscha and Fragilariopsis were on the top 20 more abundant list (Fig. 3)

Line 466: State this is the chl a-b binding protein, otherwise readers can't tell from the figure

Line 475: A really interesting result with silicon transporters solely identified in diatoms and dictyochophytes

Line 550: I thought most expressed are ribosomal proteins (top bar) (Fig. 9)

Line 532: The microscope data is not well integrated into the main text. I didn't catch them in any of the main figures. Could consider making Fig. S13 a main figure or table

Methods

Line 587: What kind of sequencing?

Line 628: What did the custom taxonomic database consist of? More details needed.

Line 706: More details are needed to reproduce the PLS analysis

Figures

Fig. 1: Are a-c and d-e showing the same exact data in two different ways? If not, I don't follow how they are different. If yes, would be more straight forward to choose 1 orientation.

Fig. 3: I have trouble interpreting the correlation plot results. Is the perpendicular orientation of diatoms and nutrient/DIN ratios really indicative of negative relationships? The negative correlation with temperature is clear, and positive relationships with chl/NO₃ are also clear. Why are the nutrients normalized to DIN? It would be interesting to see whether diatoms correlate with bulk concentration of these nutrients.

Fig. 4C: If I am understanding correctly, each dot is a different size fraction with the breakdown of diatom groups shown. It seems that size fraction is a big component of this and should be part of the visualization, for example to show that raphid pennates were more abundant in the small size fractions (captured in Fig. S8). Perhaps make each size fraction a shape (four different shape options total), so that this information can also be included here in the main text.

Fig. 9: I am not convinced this needs to be a figure in the main text. If this is showing PFam counts relative to total eukaryotic read counts, how is this reflective of diatoms? It would be more informative to subset out diatom PFam metatranscriptomic reads and take a look at how functional composition differed across sites, size fractions, and lineages (raphid, araphid, etc.).

Fig. 10 Is more useful for the functional analysis with the interesting LHC patterns with depth and size fractions, but I wonder if this would be more effective as maps instead of box and whiskers.

(Remarks on code availability)

I briefly looked through the Gitlab page and note the code is organized by amplicon analysis (V9 and V4), but note that I don't have direct experience yet working with amplicon bioinformatic processing.

Reviewer #2

(Remarks to the Author)

Karlusich et al present a global study that assesses the ecological patterns of marine diatoms, and factors driving their distribution patterns. While marine diatom diversity and ecology have been extensively studied in previous studies, this study's strength is the considerably larger sampling effort, in particular, the inclusion of the Arctic Ocean. The authors use V4 and V9 metabarcoding data from Tara Oceans supplemented by metagenomics, meta-transcriptomics, and microscopy data to show that: (1) diatoms are highly abundant in the polar regions and this abundance is linked to nitrate availability; moreover, the Arctic Ocean has distinct diatom communities (2) up to 25 sub-communities of diatoms could be detected, each with different biogeographies and factors driving distribution; (3) polar diatoms upregulate the gene expression of certain genes such as cold-shock proteins and photoprotection proteins, while those at lower latitudes express heat-shock proteins, indicating the link between genetic adaptations and environmental factors.

I think the ms is well-written, the analyses are comprehensive and appropriate, and generally well-described. The authors provide all the raw, and processed data, which I believe will be of interest to other researchers. I have only a few suggestions that the authors might want to consider for improving their ms.

1. I did not see a Code Availability statement. Moreover, the code required to go from the processed metabarcoding data to the ecological analyses and generating the plots is missing.

Minor comments:

1. Line 69. Add comma after "cell wall"
2. Line 94. Missing bracket after "metabarcoding"
3. Line 178-183. I didn't quite follow the reasoning here. Could it not simply be that 20 million reads are enough to capture the global diatom diversity?
4. Line 301. "Distribution" or "ecological" patterns of the most abundant diatom genera
5. Line 343. I did not find Supplementary File 1.
6. Line 466. The text refers to LHC proteins while the corresponding figure (Fig 9) refers to chlorophyll A-B binding proteins. The different terminologies were confusing.
7. Line 607. The link does not work and requires a sign-in.
8. Figure 1. Please consider using panel headings to make the figure easier to read.
9. Figure 3a. Consider relabeling "diatoms" as "diatom abundance".
10. Figure 7 caption. Line 854. Weird phrasing – "each V9 and V4 datasets".
11. Figure 7 caption. Line 863. Remove "and covering".
12. Figure 8b. Please write out the physiochemical variables on y axis. For instance, what is acCDOM?
13. Line 867. Clustering misspelt.
14. Figure 10. Panels f and g do not match. Which are the light harvesting proteins and which are the photo-protection ones?

(Remarks on code availability)

The repository includes code for processing metabarcoding data to generating ASV tables. However, the scripts used for the ecological analyses and generating the figures are missing.

Version 1:

Reviewer comments:

Reviewer #1

(Remarks to the Author)

The manuscript is much improved and recommended for publication.

Can the authors please ensure that the data availability links are updated to include the new sites/regions incorporated into this study, and not only the original Tara Oceans dataset from 2009-2013. For example this link hasn't been updated since 2017: <https://doi.pangaea.de/10.1594/PANGAEA.875582>. Maybe the sites including in this manuscript are already here, but they were not included in the original analysis? If this is the case, please specifically state this so other users can easily find and access these polar samples. Do the ASV tables on Zenodo also include the new sites? Ex: <https://zenodo.org/records/13881376>.

For Fig. 3A, the correlation coefficients in Supplemental Table 3 do not match the color scale bar shown. I suggest its own color bar updated to the scale of the coefficients in Table S3.

(Remarks on code availability)

REVIEWER COMMENTS

Reviewer #1 (Remarks to the Author):

“Patterns and drivers of diatom diversity and abundance in the global ocean” by Karlusich et al. is a follow up to Malviya et al. 2016 (PNAS) in which diatom diversity across the global ocean was first presented. This current study aims to address underlying factors driving the abundance and functional diversity of marine diatoms by leveraging a more exhaustive global Tara Oceans data set (double the number of sites, includes a polar region, which was missed in the original analysis). They dig further into taxonomic diversity through amplicon sequences as well as characterize functional diversity through metatranscriptomics. The comparison of V4 and V9 18S rRNA amplicon sequencing results throughout the manuscripts is especially valuable from a methodological standpoint, as there is little consistency in the field in terms of which amplicon region is being applied to survey phytoplankton communities. These results are convincing that both amplicon regions yield similar results for the diatoms.

This study is able to achieve high level insights into diatom ecology and evolution that is only possible through the compilation and analysis of global data, and the results are noteworthy. The abundance and diversity correlations with other groups is especially interesting, including the negative relationship with cyanobacteria and radiolarians, and positive relationships with copepods. The biogeographical trends among lineages of diatoms is also insightful, as it provides information their distinct evolutionary trajectories (e.g., araphid pennates in sea ice).

I have several comments regarding normalization approaches and the MAG analysis that should be addressed by the authors. However I don't think this invalidates the study, and I recommend this manuscript for publication following revisions. Detailed comments are provided below.

REPLY: We are grateful to the reviewer for her/his encouraging comments. In the following lines we offer a reply to her/his helpful suggestions.

I am wondering about normalization strategies within the V9 Tara datasets, and whether differences in community biomass should be accounted for. For example in Fig. 7, if the objective is to determine where these 20 taxa are relatively abundant and where they dominate the diatom community, wouldn't it make more sense to look at their read counts relative to total diatom reads? Diatoms are generally abundant in the Southern Ocean, so this might be why many of them appear elevated in this location when normalized to % eukaryotic reads. A few sentences about how the normalization strategy was chosen and any potential caveats would be helpful for readers to interpret biological trends.

REPLY: We thank the reviewer for this important observation. We have now included a comparison of both normalization strategies in Fig. 6a and provided further details in the text (lines 356-388 in the tracked change version). Our analysis highlights a trade-off to define the top 20 most abundant genera: normalizing by total eukaryotic reads tended to overemphasize taxa in high-latitude regions dominated by diatoms (*Fragilaria*, *Stellarima*, and *Attheya*), while normalizing by total diatom reads overemphasized taxa in temperate waters where overall diatom abundance was lower (*Haslea*, *Coscinodiscus*, *Cyclotella*, and *Pleurosigma*). Despite these biases, the majority of genera appear

consistently in the top 20 list across both normalization approaches. We opted to present both normalization strategies to allow readers to see which taxa were consistently abundant versus those whose prominence was influenced by the normalization method.

The reported increase in LHC read copy numbers at high latitude is not obvious from Fig. 10h. I recommend either showing statistical support, or removing this metagenomic component from the text. It's highlighted in the abstract and conclusion so it is a strong selling point at the moment that I am not sure is supported by the analysis.

REPLY: We appreciate the opportunity to clarify our work. We agree that the data currently presented does not provide sufficient support for an increase in LHC read copy numbers at high latitudes. As a result, we have removed the corresponding sentence from the abstract, and we have amended the results and conclusions sections to emphasize the need for more data, specifically additional MAGs with higher completeness, to properly evaluate this observation (lines 715-721 and 763-765 in the tracked change version). This remains a challenge, particularly for eukaryotic organisms.

Detailed comments

Line 142: "Second most occurring group" – I don't see what is meant by this or which figure shows this

REPLY: We recognize that the phrase was unclear, and have therefore remove it (line 149 in the tracked change version).

Line 152 and again in conclusion on line 541: In what way could diatoms be used as a bioindicator of ocean health for monitoring purposes. Is a very low relative abundance of diatoms an "unhealthy" ecosystem? More explanation is needed.

REPLY: We acknowledge that the term 'health' was ambiguous so have decided to remove it (line 159 in the tracked change version). Monitoring diatom communities is more accurately described as assessing deviations from an equilibrium or reference state. Given their critical role in ecosystem processes (such as energy flow, carbon sequestration, and resource availability) and their sensitivity to environmental changes, diatom abundance can serve as a valuable proxy for detecting early signals of broader ecosystem changes (shifts in resource balance or rising stressors, like acidification), that could otherwise be challenging to monitor. Additionally, diatoms are well-documented indicators of pollution and eutrophication across various aquatic ecosystems, including marine environments.

Line 160: This study has the light microscopy data to comment on whether their high relative abundance in 18S amplicons is likely related to having a high 18S copy number (and large biovolume). Are they also relatively abundant in image data?

REPLY: The comparison between the gene markers 18S rRNA (multicopy) and *psbO* (single-copy) relative to microscopy data is depicted in Figure 2c of Pierella Karlusich et al. (2023) *Molecular Ecology Resources* 23(1), 16-40. This figure illustrates the percentage representation of each major group within the eukaryotic phytoplankton community. The data indicate that in the 5-20 µm size fraction, diatoms are overrepresented in 18S rDNA metabarcoding data, whereas, in the 20-180 µm size fraction—a more traditional size grouping—the opposite trend is observed. This work is cited in the main text (reference 41 in line 162 in the tracked change version).

Line 176: Could the similar number of ASVs also be related to V9 not being able to differentiate among strains (multiple genera/species share the entire region), while the longer but less deeply sequenced V4 can?

REPLY: We agree with the author and have added the corresponding information (line 185-192 in the tracked change version).

Line 213: These correlation results are really interesting, especially the concepts around competition for N with cyanobacteria and for Si with radiolarians. Can PLS correlation metrics be included in the text to quantify the strength of these relationships? Even just knowing which are very strong vs. moderate would be helpful.

REPLY: We appreciate the reviewer's insightful comment regarding competition for nutrients between diatoms and other plankton. We have emphasized the corresponding correlation results more prominently in the conclusions section (lines 740-743 in the tracked change version). To address the request for quantification of these relationships, we have added the PLS coefficient scores as Supplementary Tables S2 and S3. Furthermore, we have now included a plot of the Spearman correlations for each variable in figure 3a. This addition complements the PLS analysis, which has fewer samples due to missing values in some variables, thereby affecting the overall sample size.

Line 235: What about filtered biomass aggregating on filters and resulting in smaller porosity? This could be how small cells end up on large filters.

REPLY: Indeed. This possibility has been added (line 280 in the tracked change version).

Line 438: "Pennate diatoms.. only occasionally found in the plankton" – this doesn't make sense given *Pseudo-nitzscha* and *Fragilariopsis* were on the top 20 more abundant list (Fig. 3)

REPLY: We have reworded the phrase (lines 568-573 in the tracked change version).

Line 466: State this is the chl a-b binding protein, otherwise readers can't tell from the figure

REPLY: Thank you for pointing this out. We have updated the text and figure to consistently refer to this family as light-harvesting chlorophyll a/b-binding complex (LHC) proteins to avoid confusion (lines 621-624 in the tracked change version).

Line 475: A really interesting result with silicon transporters solely identified in diatoms and dictyochophytes

REPLY: Thank you for this comment.

Line 550: I thought most expressed are ribosomal proteins (top bar) (Fig. 9)

REPLY: Thank you for this comment. If we plot strictly single Pfam families, LHCs are the most abundant. However, to reduce redundancy, we merged the dozens of Pfam families for ribosomal proteins and ubiquitin domains, which resulted in them ranking higher than LHCs. We have reworded the text to clarify this point (lines 616-624 in the tracked change version).

Line 532: The microscope data is not well integrated into the main text. I didn't catch them in any of the main figures. Could consider making Fig. S13 a main figure or table

REPLY: Thank you for highlighting this point. We have now incorporated the microscopy data in Fig 6b.

Methods

Line 587: What kind of sequencing?

REPLY: We replaced 'high-throughput' by 'Illumina' (line 798 in the tracked change version).

Line 628: What did the custom taxonomic database consist of? More details needed.

REPLY: This information has now been added (lines 842-846 in the tracked change version).

Line 706: More details are needed to reproduce the PLS analysis

REPLY: Reply: Additional details regarding the PLS analysis have been included in the revised manuscript (lines 936-942 in the tracked change version). Furthermore, the code used for the analysis is publicly available on GitHub at

https://github.com/JJPierellaKarlusich/Diatom_patterns/blob/main/metaB/Fig3a_abiotic.R and https://github.com/JJPierellaKarlusich/Diatom_patterns/blob/main/metaB/Fig3a_biotic.R

Figures

Fig. 1: Are a-c and d-e showing the same exact data in two different ways? If not, I don't follow how they are different. If yes, would be more straight forward to choose 1 orientation.

REPLY: Thank you for your comment. We have added panel headings to make the distinctions between the panels clearer.

Fig. 3: I have trouble interpreting the correlation plot results. Is the perpendicular orientation of diatoms and nutrient/DIN ratios really indicative of negative relationships? The negative correlation with temperature is clear, and positive relationships with chl/NO₃ are also clear. Why are the nutrients normalized to DIN? It would be interesting to see whether diatoms correlate with bulk concentration of these nutrients.

REPLY: We now replaced nutrient/DIN ratios by the bulk nutrient concentrations in Fig3 and all related figures throughout the manuscript. We decided to keep the ratio NH₄⁺/DIN, as there is evidence that the relative availability and/or supply of NH₄⁺ compared to the more oxidized forms of NO₂⁻ and NO₃⁻ is a modulator of the phytoplankton community (e.g., Buchanan, et al. 2024 *Enrichment of ammonium in the future ocean threatens diatom productivity. ESS Open Archive*). Additionally, we complemented the PLS analysis with Spearman's rho correlation analyses, as the PLS sample size was reduced due to the exclusion of samples with missing values for certain environmental variables.

Fig. 4C: If I am understanding correctly, each dot is a different size fraction with the breakdown of diatom groups shown. It seems that size fraction is a big component of this and should be part of the visualization, for example to show that raphid pennates were more abundant in the small size fractions (captured in Fig. S8). Perhaps make each size fraction a shape (four different shape options total), so that this information can also be included here in the main text.

REPLY: We have added an extra panel to Figures 4c and S7c where samples are now color-coded by size fraction. This new visual representation showed some partitioning by size fraction. However, the pore size of the lower filter was included as a variable in the NMDS analysis, and this variable was not statistically significant.

Fig. 9: I am not convinced this needs to be a figure in the main text. If this is showing PFam counts relative to total eukaryotic read counts, how is this reflective of diatoms? It would be more informative to subset out diatom PFam metatranscriptomic reads and take a look at how functional composition differed across sites, size fractions, and lineages (raphid, araphid, etc.).

REPLY: The diatom Pfams were normalized by total diatom reads and compared across size fractions and oceanic regions in the left panels of Figure 9. Additionally, equivalent values were calculated for other main eukaryotic phytoplankton groups to compare them with diatom patterns, and these are displayed in the right panel of Figure 9. We have reworded both the Figure 9 caption and the horizontal axis label to improve clarity (lines 1182-1203 in the tracked change version), and we propose to retain this information in the main text because we consider it to be highly pertinent to the manuscript.

Fig. 10 is more useful for the functional analysis with the interesting LHC patterns with depth and size fractions, but I wonder if this would be more effective as maps instead of box and whiskers.

REPLY: We have replaced the latitudinal gradient panel by a map for DUF285.

Reviewer #1 (Remarks on code availability):

I briefly looked through the Gitlab page and note the code is organized by amplicon analysis (V9 and V4), but note that I don't have direct experience yet working with amplicon bioinformatic processing.

REPLY: We have now added a Code Availability statement with the link to the scripts for the ecological analysis and for reproducibility of the figures (lines 1022-1027 in the tracked change version).

Reviewer #2 (Remarks to the Author):

Karlusich et al present a global study that assesses the ecological patterns of marine diatoms, and factors driving their distribution patterns. While marine diatom diversity and ecology have been extensively studied in previous studies, this study's strength is the considerably larger sampling effort, in particular, the inclusion of the Arctic Ocean. The authors use V4 and V9 metabarcoding data from Tara Oceans supplemented by metagenomics, meta-transcriptomics, and microscopy data to show that: (1) diatoms are highly abundant in the polar regions and this abundance is linked to nitrate availability; moreover, the Arctic Ocean has distinct diatom communities (2) up to 25 sub-communities of diatoms could be detected, each with different biogeographies and factors driving distribution; (3) polar diatoms upregulate the gene expression of certain genes such as cold-shock proteins and photoprotection proteins, while those at lower latitudes express heat-shock proteins, indicating the link between genetic adaptations and environmental factors.

I think the ms is well-written, the analyses are comprehensive and appropriate, and generally well-described. The authors provide all the raw, and processed data, which I believe will be of interest to other researchers. I have only a few suggestions that the authors might want to consider for improving their ms.

REPLY: We are pleased to know that the manuscript was well received, and we thank the reviewer for her/his helpful comments and corrections. We address them below.

1. I did not see a Code Availability statement. Moreover, the code required to go from the processed metabarcoding data to the ecological analyses and generating the plots is missing.
REPLY: We have now added a Code Availability statement with the link to the scripts for the ecological analysis and for reproducibility of the figures (lines 1022-1027 the tracked change version).

Minor comments:

1. Line 69. Add comma after “cell wall”

REPLY: Added (line 74 in the tracked change version).

2. Line 94. Missing bracket after “metabarcoding”

REPLY: Added (line 100 in the tracked change version).

3. Line 178-183. I didn't quite follow the reasoning here. Could it not simply be that 20 million reads are enough to capture the global diatom diversity?

REPLY: We thank the reviewer for this remark and we have now modified the text (lines 185-192 in the tracked change version).

4. Line 301. “Distribution” or “ecological” patterns of the most abundant diatom genera

REPLY: Changed to ‘Biogeography of the most abundant diatom genera’ (line 356 in the tracked change version).

5. Line 343. I did not find Supplementary File 1.

REPLY: Thank you for bringing this to our attention. Supplementary File 1 was submitted with the manuscript and is available in the tracking system of our account. There may have been an issue with the journal's website. We recommend checking with the editorial team to ensure the file is properly accessible to the reviewers. The same file can also be accessed from the previous preprint version: <https://www.biorxiv.org/content/biorxiv/early/2024/06/10/2024.06.08.598090/DC2/embed/media-2.pdf>

6. Line 466. The text refers to LHC proteins while the corresponding figure (Fig 9) refers to chlorophyll A-B binding proteins. The different terminologies were confusing.

REPLY: Thank you for pointing this out. We have updated the text and figure to consistently refer to this family as light-harvesting chlorophyll a/b-binding complex (LHC) proteins to avoid confusion (lines 621-624 in the tracked change version).

7. Line 607. The link does not work and requires a sign-in.

REPLY: Thank you for bringing this to our attention. The written link pathway was correct, but a line break caused an error that redirected to a higher-level directory. We have confirmed that the link now works properly (lines 819-820 in the tracked change version) and are also providing it here for reference: <https://gitlab.sb-roscoff.fr/nhenry/abims-metabarcoding-taxonomic-assignment/-/tree/v1.0.1>

8. Figure 1. Please consider using panel headings to make the figure easier to read.

REPLY: Thank you for the suggestion. We have added panel headings to Figure 1 to enhance clarity.

9. Figure 3a. Consider relabeling “diatoms” as “diatom abundance”.

REPLY: Done.

10. Figure 7 caption. Line 854. Weird phrasing – “each V9 and V4 datasets”.

REPLY: Rephrased (lines 1136 in the tracked change version).

11. Figure 7 caption. Line 863. Remove “and covering”.

REPLY: Corrected (line 1142 in the tracked change version).

12. Figure 8b. Please write out the physiochemical variables on y axis. For instance, what is acCDOM?

REPLY: Added (lines 1167-1180 in the tracked change version).

13. Line 867. Clustering misspelt.

REPLY: Corrected (line 1162 in the tracked change version).

14. Figure 10. Panels f and g do not match. Which are the light harvesting proteins and which are the photo-protection ones?

REPLY: Thank you for pointing this out. The figure is now corrected. LHCf, LHCq, and LHCr are the light harvesting subfamilies, and LHCx and LHCz are the photoprotecting ones.

Reviewer #2 (Remarks on code availability):

The repository includes code for processing metabarcoding data to generating ASV tables. However, the scripts used for the ecological analyses and generating the figures are missing.

REPLY: We have now added a Code Availability statement with the link to the scripts for the ecological analysis and for reproducibility of the figures (lines 1022-1027 in the tracked change version).

REVIEWERS' COMMENTS

Reviewer #1 (Remarks to the Author):

The manuscript is much improved and recommended for publication.

REPLY: We are very pleased to read this, and we thank the reviewer again for her/his helpful comments and corrections.

Can the authors please ensure that the data availability links are updated to include the new sites/regions incorporated into this study, and not only the original Tara Oceans dataset from 2009-2013. For example this link hasn't been updated since 2017: <https://doi.pangaea.de/10.1594/PANGAEA.875582> Maybe the sites including in this manuscript are already here, but they were not included in the original analysis? If this is the case, please specifically state this so other users can easily find and access these polar samples. Do the ASV tables on Zenodo also include the new sites? Ex: <https://zenodo.org/records/13881376>

REPLY: We confirm that the data availability links (Pangaea contextual data, ASV tables on Zenodo, and metatranscriptomic tables on Genoscope website) were already updated and include all new sites analyzed in this study, beyond the transect between 09/2009 and 08/2011 analysed by Malviya et al 2016 PNAS. The new release datasets included those generated from samples collected during the Arctic circumnavigation in 2013. We have added a clear statement in the Methods and Data Availability sections to ensure users can easily locate and access these updates.

For Fig. 3A, the correlation coefficients in Supplemental Table 3 do not match the color scale bar shown. I suggest its own color bar updated to the scale of the coefficients in Table S3.

REPLY: The correlation coefficients in Supplementary Table 3 correspond to those from the PLS analysis, while the color bar in Figure 3A represents Spearman's rho correlation analysis. We have modified the main text and the figure caption to explicitly state this distinction.